# RIF1-ASF1-mediated high-order chromatin structure safeguards genome integrity

Sumin Feng[1,4], Sai Ma[1,4], Kejiao Li[1,4], Shengxian Gao[1], Shaokai Ning[1], Jinfeng Shang[1], Ruiyuan Guo[1], Yingying Chen[2], Britny Blumenfeld[3], Itamar Simon[3], Qing Li [1,2], Rong Guo [1] & Dongyi Xu [1✉]

The 53BP1-RIF1 pathway antagonizes resection of DNA broken ends and confers PARP inhibitor sensitivity on BRCA1-mutated tumors. However, it is unclear how this pathway suppresses initiation of resection. Here, we identify ASF1 as a partner of RIF1 via an interacting manner similar to its interactions with histone chaperones CAF-1 and HIRA. ASF1 is recruited to distal chromatin flanking DNA breaks by 53BP1-RIF1 and promotes non-homologous end joining (NHEJ) using its histone chaperone activity. Epistasis analysis shows that ASF1 acts in the same NHEJ pathway as RIF1, but via a parallel pathway with the shieldin complex, which suppresses resection after initiation. Moreover, defects in end resection and homologous recombination (HR) in BRCA1-deficient cells are largely suppressed by ASF1 deficiency. Mechanistically, ASF1 compacts adjacent chromatin by heterochromatinization to protect broken DNA ends from BRCA1-mediated resection. Taken together, our findings identify a RIF1-ASF1 histone chaperone complex that promotes changes in high-order chromatin structure to stimulate the NHEJ pathway for DSB repair.

[1] State Key Laboratory of Protein and Plant Gene Research, School of Life Sciences, Peking University, 100871 Beijing, China. [2] Peking-Tsinghua Center for Life Sciences, Academy for Advanced Interdisciplinary Studies, Peking University, 100871 Beijing, China. [3] Department of Microbiology and Molecular Genetics, Institute of Medical Research Israel-Canada, Faculty of Medicine, The Hebrew University, Jerusalem 91120, Israel. [4] These authors contributed equally: Sumin Feng, Sai Ma, Kejiao Li. ✉email: xudongyi@pku.edu.cn

Double-strand breaks (DSBs) are one of the most cytotoxic DNA lesions and must be effectively and accurately repaired to prevent genomic instability, carcinogenesis, and cell death. To this end, cells must properly choose from two mutually exclusive major DSB repair pathways, homologous recombination (HR) and non-homologous end joining (NHEJ), based on cell cycle position and the nature of the DNA end. DNA end resection, in which broken DNA is converted into 3′-overhang ends suitable for HR, plays a central role in determining the DSB repair pathway choice and is controlled by functional antagonism between the HR-promoting factor BRCA1 and NHEJ-promoting proteins 53BP1 and RIF1[1–7]. Mutations in *BRCA1* or *BRCA2* genes cause breast, ovarian, prostate, and other cancers, and tumors with such mutations show PARPi hypersensitivity[8,9]. Unfortunately, PARPi resistance is frequently acquired in patients with advanced cancer. In addition to the restoration of BRCA1/2 expression or function by secondary mutations, loss of 53BP1 or its downstream effectors is one major mechanism of PARPi resistance[10].

End resection is initiated at proximal chromatin flanking DSBs by BRCA1-promoted CtIP-MRN, and is extended by EXO1 and DNA2/BLM. Recent studies reveal that the shieldin complex (REV7-SHLD1-SHLD2-SHLD3), which is downstream of 53BP1-RIF1, counteracts end resection through CST- and Polα-dependent fill-in[11–20]. As the shieldin-CST-Polα pathway acts on single-strand DNA (ssDNA)[13,16,17,20], this pathway acts like a retrieval system after resection is initiated by mistakes (or in a "trial-and-error" way between HR and NHEJ), or as a restriction system to limit over-resection. In principle, it's more critical for NHEJ to suppress endonuclease-mediated resection on chromatin flanking DSBs at the initiation step in comparison with the extension step. However, whether and how this function is executed by the 53BP1 pathway remains unclear, although it has long been postulated that the 53BP1-RIF1 complex strengthens the nucleosomal barrier to end-resection nucleases[16,21,22].

ASF1 is a histone chaperone conserved from yeast to human cells. Higher eukaryotes contain two paralogs of yeast ASF1, ASF1a and ASF1b, which are distinguishable by their C-terminal tails[23]. ASF1 functions by transferring H3–H4 heterodimers to the histone chaperone CAF-1 or HIRA for nucleosome assembly[23] and contributes to heterochromatin formation[24–26]. In addition to its role in nucleosome assembly, ASF1 also plays a role in nucleosome disassembly and histone exchange[27–30]. Here, we find that ASF1 forms a complex with RIF1 in response to DNA damage through a B-domain, which is also responsible for the interactions of CAF-1 and HIRA with ASF1. ASF1 promotes formation of high-order chromatin structure, antagonizes BRCA1-dependent DNA end resection and stimulates NHEJ via its histone chaperone activity. Thus, we identify a RIF1-ASF1 histone chaperone complex that protects broken DNA ends in a parallel pathway with shieldin and confers PARPi sensitivity on BRCA1-deficient cells.

## Results

### The shieldin complex does not promote NHEJ as effectively as RIF1.
Previously, we and other groups identified the shieldin complex as a downstream effector of RIF1 to antagonize BRCA1-medicated HR and promote NHEJ[13–19]. Cell survival experiments using a MTT assay showed that *rif1*−/− cells were more sensitive to ICRF193, a topoisomerase II inhibitor that induces DSBs and is specifically toxic to cells deficient in NHEJ[31–33], in comparison with SHLD2/FAM35A-deficient cells (see Fig. 1b in ref. [13]). We confirmed this result using a colony-formation assay with sensitivity greater than that of the MTT assay (Supplementary Fig. 1a). PARPis induce one-end DSBs during replication and are toxic to cells deficient in HR proteins such as BRCA1. Disruption of the 53BP1-RIF1 pathway restores HR in BRCA1-deficient cells and

thus reduces their PARPi sensitivity[1–7]. Consistent with the results of the experiments assessing ICRF193 sensitivity, disruption of SHLD2 was not as effective as knockout of RIF1 with regard to rescuing the PARPi (olaparib) sensitivity of *brca1*−/− cells (Supplementary Fig. 1b, c). Therefore, the shieldin complex only plays a partial role in mediating antagonism of BRCA1 and the promotion of NHEJ by RIF1.

In addition, the colony-formation assay showed that *shld2*−/− cells were more sensitive to etoposide, another topoisomerase II inhibitor that induces DSBs that can be repaired by NHEJ or other pathways[31,32], in comparison with *rif1*−/− cells (Supplementary Fig. 1a), suggesting that the shieldin complex may have more functions in addition to promoting NHEJ to repair DSBs.

### RIF1 forms a histone chaperone complex with ASF1.
To explore the potential parallel pathways of the shieldin complex, we immunopurified and analyzed the RIF1 complexes from HEK293 cells transiently expressing FLAG-RIF1 with an anti-FLAG antibody. Mass spectrometry analysis revealed that the histone chaperone protein, ASF1a, and H3–H4 were co-immunoprecipitated with RIF1 (Fig. 1a and Supplementary Data 1), and immunoblotting confirmed this finding (Fig. 1b). Immunoblotting and mass spectrometry analysis of reciprocal immunoprecipitation showed that RIF1 was present as one major component in the immunoprecipitate of FLAG-ASF1a (Fig. 1a, c), which is consistent with previous interactome analyses[34–36]. These results demonstrate that RIF1 forms a stable complex with ASF1a-H3–H4. RIF1 interacts with 53BP1 to protect DNA broken ends[4–7]. 53BP1 was detected in the FLAG-ASF1a immunoprecipitate by immunoblotting, although it was not identified by mass spectrometry (Fig. 1a, c), suggesting that this complex associates with 53BP1 possibly to protect DSB ends. RIF1 and 53BP1 accumulated more in the immunoprecipitates of FLAG-ASF1a from the chromatin fraction in comparison with the soluble fraction (Supplementary Fig. 2a), indicating that this complex mainly forms on chromatin.

Moreover, the interaction of ASF1a with RIF1, but not HIRA or CAF-1, was enhanced by DNA damage induced by bleomycin (Fig. 1b, c and Supplementary Fig. 2b, c), implying that RIF1-ASF1a may play a role in the response to DNA damage. ASF1-bound non-nucleosomal H3–H4 heterodimers contain pre-modified H3K9me1, particularly in genotoxic conditions[37]. Interestingly, H3K9me1 was enriched in the RIF1–ASF1 complex when DNA was damaged (Fig. 1b), implying that RIF1-bound ASF1 tends to provide H3K9me1 upon DNA damage.

ASF1b, a paralog of ASF1a, was not enriched as effectively as ASF1a in the FLAG-RIF1 immunoprecipitate (Fig. 1b), although RIF1 was present as one major component in the FLAG-ASF1b immunoprecipitate (Fig. 1a, c), suggesting that only a subset of RIF1 forms complex with ASF1b.

The downstream histone chaperones CAF-1 p60 and HIRA interact with ASF1 in a mutually exclusive manner via a B-domain motif[38,39]. Sequence alignment analysis reveals that RIF1 contains a putative B-domain with high similarity to the domains present in CAF-1 p60, HIRA, and CDAN1 (Fig. 1d). Mapping the interacting region revealed that a conserved region (aa1180–1270) of RIF1, which contains the B-domain, was necessary and sufficient for its interaction with ASF1a (Supplementary Fig. 2d–f). Importantly, point mutation in the B-domain of RIF1 (R1217A/R1218A/Q1219D) dramatically reduced its interaction with ASF1a (Fig. 1d, e and Supplementary Fig. 2d). Consistently, mutation of the ASF1a residues (E36A/D37A) critical for binding the B-domain in CAF-1 and HIRA, but not the residue (V94R) for binding H3–H4, disrupted its interaction with RIF1 (Supplementary Fig. 2g, h). Taken together, these results demonstrate that ASF1a binds RIF1 in a manner similar to

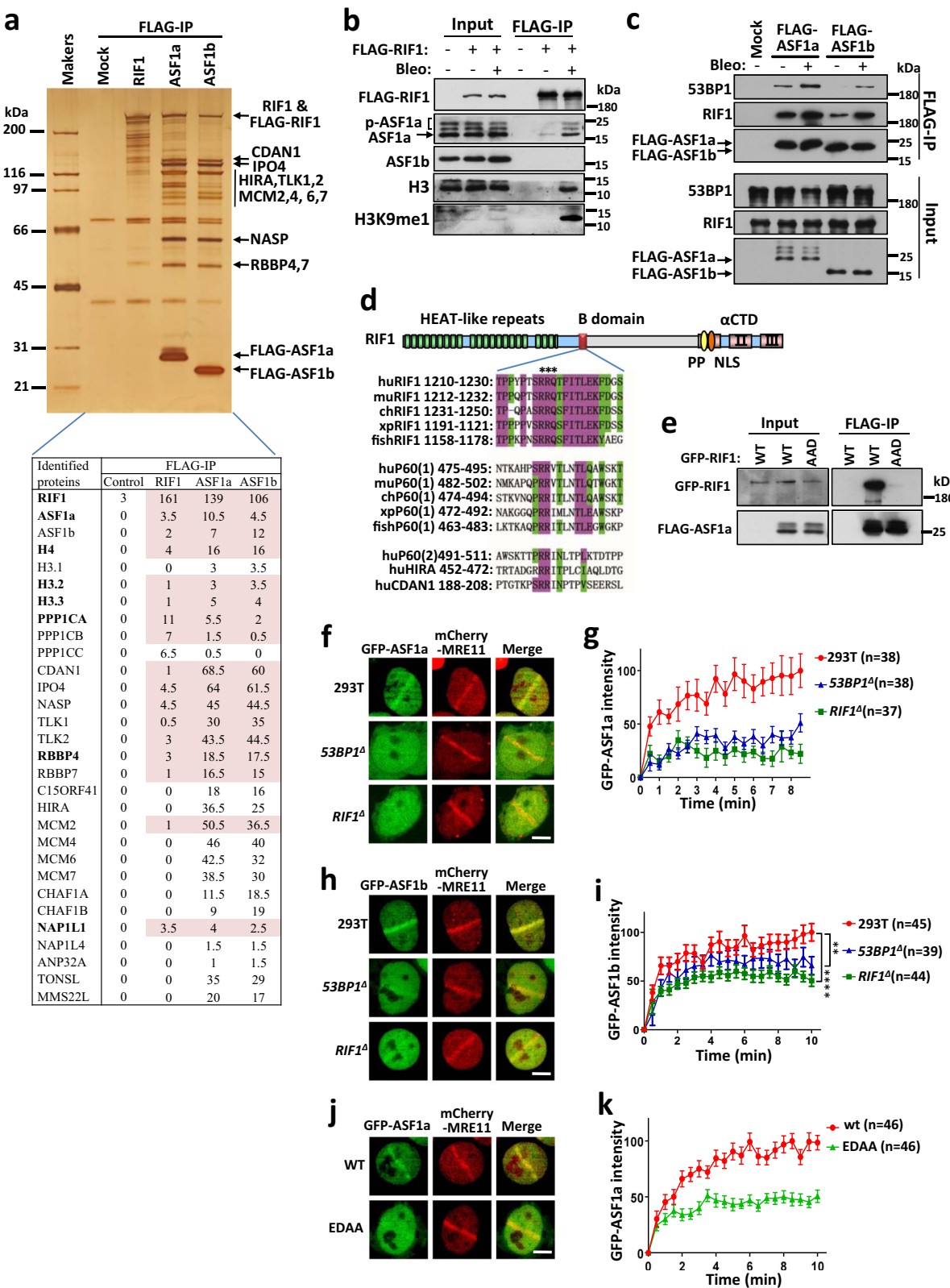

its interaction with CAF-1 and HIRA, implying that these interactions are mutually exclusive.

**ASF1 is recruited to DSB sites by 53BP1-RIF1.** We observed that both GFP-tagged ASF1a and ASF1b were recruited to sites of laser-induced DNA damage (Fig. 1f–i). Recruitment of GFP-ASF1a was dramatically decreased when 53BP1 or RIF1 was

disrupted (Fig. 1f, g and Supplementary Fig. 3a), demonstrating that ASF1a is mainly recruited to DNA damage sites by 53BP1-RIF1. This result was further confirmed by endogenous ASF1a staining (Supplementary Fig. 3b-d). The recruitment of GFP-ASF1b was only modestly reduced in 53BP1- or RIF1-null cells (Fig. 1h, i), suggesting that proteins other than 53BP1-RIF1 also contribute to its recruitment to DSB sites. Consistently, ASF1a

**Fig. 1 ASF1 forms a complex with RIF1 and is recruited to DSB sites by 53BP1-RIF1. a** A silver-stained SDS-PAGE gel showing the polypeptides that were immunopurified from extracts of HEK293 cells expressing FLAG-tagged RIF1, ASF1a, and ASF1b using the anti-FLAG antibody. HEK293 cells without expressing exogenous protein were included as a control (mock). The major polypeptides on the gel were identified by mass spectrometry. The mean numbers of peptides discovered for each protein from two replicates are listed at the bottom. A full list of mass spectrometry data is available in Supplementary Data 1. **b, c** Immunoblot showing IP of FLAG-tagged RIF1 (**b**), ASF1a, and ASF1b (**c**). HEK293 cells expressing FLAG-RIF1 were treated with/ without bleomycin (20 μg/mL) for 3 h before harvest. **d** Schematic representation of RIF1 and sequence alignment of the B domains of RIF1, CAF-1 P60, HIRA, and CDAN1. For proteins with two motifs, both are aligned and designated (1) and (2). hu *Homo sapiens*, mu *Mus musculus*, ch *Gallus gallus*, xp *Xenopus laevis*, fish *Danio rerio*. **e** Immunoblot showing FLAG-IP from extracts of HEK293 cells expressing FLAG-tagged ASF1a and GFP-RIF1 wild-type (WT) or the AAD mutant. **f–i** Recruitment of GFP-ASF1a (**f**) and GFP-ASF1b (**h**) to laser-induced DNA damage sites in wild-type, *53BP1*$^Δ$ or *RIF1*$^Δ$ HEK293T cells and quantification (**g, i**). Mean ± s.e.m. is shown for every time point. The numbers of cells pooled from two independent experiments are indicated. **\*\***$P < 0.01$; **\*\*\*\***$P < 0.0001$; statistical analysis was performed using two-way ANOVA. **j, k** Recruitment of GFP-tagged wild-type or mutated ASF1a to laser-induced DNA damage sites (**j**) and quantification (**k**). Mean ± s.e.m. is shown for every time point. The numbers of cells pooled from two independent experiments are indicated. Representative images were taken 10 min after laser irradiation. Scale bar, 5 μm. Uncropped images of the gels and statistical source data including the precise *P* values are provided in Source data.

mutant EDAA (E36A/D37A), in which residues required for interaction with RIF1 were mutated, showed strongly reduced recruitment to DSB sites (Fig. 1j, k).

**ASF1 colocalizes with 53BP1 and RIF1 at distal chromatin flanking DSBs**. 53BP1 is excluded from chromatin proximal to DSBs by BRCA1 in the S/G2 phase, resulting in a specific spatial distribution, in which recruitment of BRCA1 into the core of the damage focus is associated with exclusion of 53BP1 to the focal periphery[40,41]. This distribution of 53BP1 at distal chromatin during HR was deemed to prevent hyper-resection and subsequent single-strand annealing (SSA) to foster its fidelity[42]. However, due to technical barriers, this spatial distribution has been observed for only a limited number of DSB repair proteins[40,41]. To examine whether other proteins downstream of 53BP1 are also recruited to chromatin distal to DSB sites, we developed a method: high-energy-laser-induced recruitment to distal chromatin flanking DSBs (HIRDC). The high-energy-laser-microirradiated regions (1–2-μm-diameter dots) in the cell nucleus are supposed to contain DSB ends, ssDNA and DSB-flanking proximal chromatin, but to largely exclude distal chromatin because of dense DSBs (Fig. 2a). Consistently, ssDNA binding proteins RAD51 and RPA were exclusively recruited in the irradiated dots (Supplementary Fig. 4a), whereas γH2AX localized both inside and outside the dots, although the inside signal was weaker (Supplementary Fig. 4a); BRCA1 and MRE11 were located in regions slightly larger than the dots marked by RPA (Supplementary Fig. 4a), in agreement with reports that BRCA1 and MRE11 are recruited to both ssDNA and proximal chromatin[43]; Interestingly, both endogenous 53BP1 and GFP-tagged 53BP1 were excluded from the irradiated dots and formed a ring surrounding BRCA1/MRE11 (Fig. 2b, c and Supplementary Fig. 4a–c). The spatial distribution of these proteins was not due to destruction of chromatin by laser microirradiation because staining for H3 and DNA was normal in these regions (Supplementary Fig. 4a). As expected, exclusion of 53BP1 was dose-dependent and impaired when BRCA1 was absent (Supplementary Fig. 4b-d), suggesting that BRCA1 promotes exclusion of 53BP1 from the core region during resection, in agreement with previous studies[40,41]. Both endogenous and GFP-tagged RIF1 showed re-localization patterns similar to that of 53BP1 (Fig. 2d, e), suggesting that they may work together to prevent hyper-resection during HR in a role similar to their function in promoting NHEJ.

Both GFP-tagged ASF1a and ASF1b were able to re-distribute at distal chromatin flanking DSBs (Fig. 2f, g). In *53BP1*$^Δ$ and *RIF1*$^Δ$ cells, these distributions of ASF1a and ASF1b were dramatically and modestly decreased, respectively (Fig. 2h–j), demonstrating that ASF1 acts downstream of 53BP1-RIF1, possibly to limit resection. Mutation in the B-domain of RIF1 significantly impaired this distribution of GFP-ASF1a at distal chromatin (Fig. 2k and

Supplementary Fig. 5a, b), suggesting that the interaction of ASF1a with RIF1 mediates this re-distribution.

Different from 53BP1 and RIF1, ASF1a and ASF1b were also recruited to the core irradiated regions, although to a lesser degree in comparison with the periphery (Fig. 2f, g). Recruitment of ASF1a and ASF1b to the core regions was not reduced by disruption of 53BP1 or RIF1 (Fig. 2h–j). These results suggest that ASF1 is also recruited to ssDNA or proximal chromatin by other proteins, consistent with the recent report that ASF1 promotes MMS22L-TONSL-mediated RAD51 loading onto ssDNA during HR[44].

**ASF1 promotes NHEJ via the same pathway as RIF1 through its histone chaperone activity**. Although HEK293T cells lacking either ASF1a or ASF1b were viable, double-knockout cells were not available, possibly due to lethality (Supplementary Fig. 6a). We generated ASF1b heterozygous mutants in *ASF1a*-null cells, which showed more than 50% reduced abundance of ASF1b (Supplementary Fig. 6a). Both *ASF1a*$^Δ$ and *ASF1b*$^Δ$ cells were sensitive to ionizing radiation (IR), while *ASF1a*$^Δ$*ASF1b*$^H$ cells were more sensitive to IR in comparison with the single knockout cells (Supplementary Fig. 6b), suggesting that the two paralogs of ASF1 have overlapping functions in DSB repair.

Chicken DT40 cells have only one paralog of ASF1, ASF1a, which is essential[45]. By fusing an auxin-inducible degron (AID) at its C-terminus, we generated ASF1a conditional knockout DT40 cells (*asf1a*$^{-/-/AID}$), which showed nearly full loss of the ASF1a protein and proliferation when auxin was present (Supplementary Fig. 6c-e). Even without auxin, *asf1a*$^{-/-/AID}$ cells, which retained <10% of the ASF1a protein level of normal cells, were sensitive to etoposide and ICRF193 (Fig. 3a and Supplementary Fig. 6d). In fact, *asf1a*$^{-/-/+}$ cells, which retained 33% of the ASF1a protein level of normal cells, also showed modest etoposide and ICRF193 sensitivity (Fig. 3a and Supplementary Fig. 6d). These results suggest that ASF1 is important for DSB repair. In contrast, both *asf1a*$^{-/-/+}$ and *asf1a*$^{-/-/AID}$ cells were not (or only very weakly at low doses) sensitive to topoisomerase I inhibitor, camptothecin (CPT; Supplementary Fig. 6f), which induces one-end DSBs that depend on the HR pathway for repair[13,31], suggesting that ASF1 is almost dispensable for HR in DT40 cells. NHEJ is essential for foreign DNA random integration (approximate 300-fold reduction in *ku70*$^{-/-}$ cells compared to wild-type DT40 cells)[4,13,46]. Random integration in *asf1a*$^{-/-/+}$ and *asf1a*$^{-/-/AID}$ cells was decreased by 3.1-fold and 6.6-fold, respectively (Fig. 3b). Together, these results demonstrate that ASF1 promotes the NHEJ pathway for DSB repair.

We performed genetic interaction analysis to determine whether ASF1 acts via the same NHEJ pathway through which RIF1 functions. The *rif1*$^{-/-}$*asf1a*$^{-/-/+}$ cells did not show more sensitivity to etoposide or ICRF193, or reduction of foreign DNA

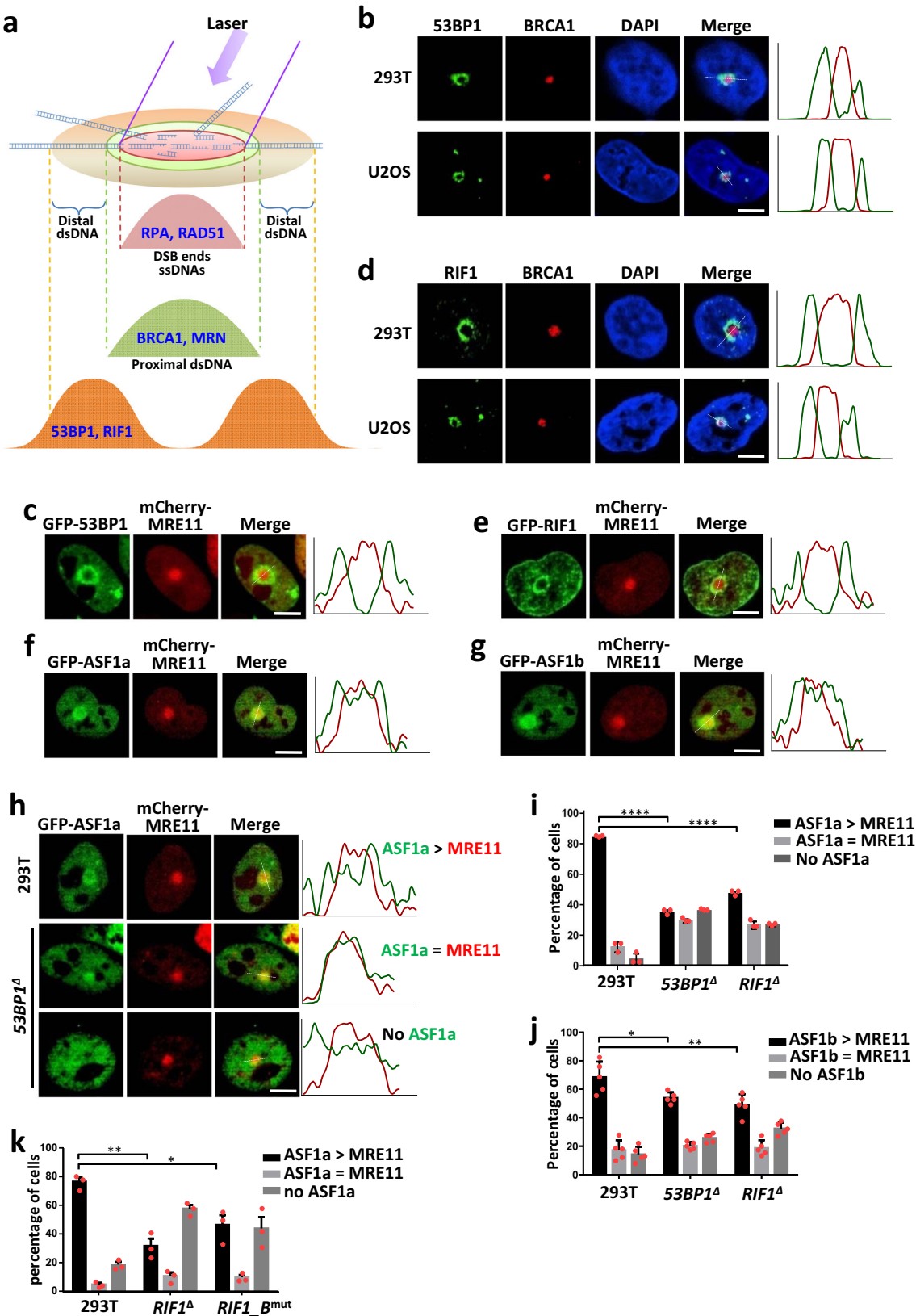

random integration, in comparison with the corresponding single knockout cells (Fig. 3a, b), demonstrating that ASF1a and RIF1 are epistatic in the NHEJ pathway. Consistently, the interaction between RIF1 and ASF1a was required for the promotion of NHEJ. The ASF1a-interacting-defective mutant (AAD) of RIF1 was not as effective as its wild-type form with regard to rescuing

the defect of $rif1^{-/-}$ cells in etoposide resistance and foreign DNA random integration (Fig. 3c, d and Supplementary Fig. 6g); wild-type ASF1a, but not its RIF1-interacting-defective mutant (EDAA), rescued the etoposide sensitivity and random integration defect of $asf1a^{-/-/AID}$ cells (Fig. 3e, f and Supplementary Fig. 6h). Thus, ASF1 and RIF1 act in the same pathway to promote NHEJ.

**Fig. 2 ASF1 is recruited to distal chromatin flanking DSBs by 53BP1-RIF1. a** Schematic representation of the distribution of DNA repair proteins in the HIRDC assay. **b** Immunofluorescence showing the distribution of 53BP1 and BRCA1 15 min after microirradiation using a high-energy laser in the HIRDC assay. The right panel shows the distribution of the red and green signals on the white dashed line as indicated in the left image. **c** GFP-53BP1 and mCherry-MRE11 in the HIRDC assay in HEK293T cells. **d** Immunofluorescence showing the distribution of RIF1 and BRCA1 in the HIRDC assay. **e–g** GFP-tagged RIF1 (**e**), ASF1a (**f**), and ASF1b (**g**) in the HIRDC assay in HEK293T cells. **h** GFP-ASF1a in the HIRDC assay in wild-type or *53BP1*Δ HEK293T cells. ASF1a > MRE11, the area of accumulated GFP-ASF1a is bigger than that of mCherry-MRE11; ASF1a = MRE11, the area of accumulated GFP-ASF1a is equal with that of mCherry-MRE11; no ASF1a, no accumulated GFP-ASF1a signal. **i–k** Quantification of the distributions of GFP-tagged ASF1a (**i, k**) and ASF1b (**j**) in the HIRDC assay in wild-type, *53BP1*Δ, *RIF1*Δ (**i, j**) or *RIF1_Bmut* (**k**) HEK293T cells. The bars represent the mean ± s.d.; *n* = 3 (**i**), 5 (**j**) and 3 (**k**) independent experiments; at least 50 cells were scored for every sample in one independent experiment; *P < 0.05; **P < 0.01; ****P < 0.0001; statistical analysis was performed using one-way ANOVA. Scale bar, 5 μm. Statistical source data, including the precise *P* values, are provided in the source data.

RIF1_AAD retains a partial function in resisting etoposide (Fig. 3c), possibly due to its other functions than NHEJ. In fact, RIF1 has been reported to resist replication stress, and disruption of RIF1 has much more severe phenotypes than that of 53BP1 in cells or mice[47,48]. In addition, the defect of HR in *asf1a*[−/−/AID] cells may also contribute to the etoposide sensitivity[44].

The shieldin complex acts downstream of 53BP1 and RIF1 to protect broken ends and promote NHEJ. Interestingly, *asf1a*[−/−]+*shld2*[−/−] cells showed more sensitivity to etoposide and ICRF193 in comparison with *shld2*[−/−] or *asf1a*[−/−/+] cells (Fig. 3g); and *rif1*[−/−]*asf1a*[−/−/+]*shld2*[−/−], *asf1a*[−/−/+]*shld2*[−/−] and *rif1*[−/−] showed similar sensitivity to ICRF193 (Fig. 3h). These results demonstrate that ASF1a and the shieldin complex act in two parallel pathways to promote NHEJ downstream of 53BP1-RIF1 (Fig. 3i).

Moreover, wild-type ASF1a, but not its histone-binding-defective mutant V94R, rescued the defect of *asf1a*[−/−/AID] cells in resisting etoposide and foreign DNA random integration (Fig. 3e, f and Supplementary Fig. 6h), demonstrating that the histone chaperone activity of ASF1 is required for its function in NHEJ.

**ASF1 protects broken ends and antagonizes HR in BRCA1-deficient cells.** We determined whether ASF1 antagonizes the function of BRCA1 in a manner similar to 53BP1-RIF1. As expected, depletion of ASF1 rescued the PARPi sensitivity of both BRCA1-deficient DT40 and HCT116 cells (Fig. 4a–d). RPA binds to resected ssDNA after end resection and is subsequently replaced by RAD51, which promotes strand invasion and strand exchange during HR[49]. Reduced foci of both RPA and RAD51 were recovered from BRCA1-deficient cells after ASF1 depletion (Fig. 4e, f and Supplementary Fig. 7a–c). These results were unlikely to have been caused by changes in the cell cycle (Supplementary Fig. 7d–f). Therefore, ASF1 opposes HR by limiting resection in BRCA1-deficient cells, similar to 53BP1 and RIF1[1–7].

Moreover, wild-type RIF1, but not its AAD mutant, recovered the PARPi sensitivity of *brca1*[−/−]*rif1*[−/−] cells (Fig. 4g, h), suggesting that the interaction of RIF1 with ASF1 is important for its function in suppressing BRCA1-dependent HR.

**The RIF1–ASF1 complex promotes heterochromatinization flanking DSBs.** DSBs induce ATM-dependent transcriptional silencing and chromatin condensation at regions flanking damage sites in U2OS-265 cells, which harbor approximately 200 copies of a LacO-TetO array integrated at a single site on chromosome 1p3.6[50–52]. Using the same system (Fig. 5a), we confirmed that mCherry-LacR-FokI-caused DSB induced chromatin condensation at the LacO array (Supplementary Fig. 8a–c), while an ATM inhibitor suppressed this condensation (Supplementary Fig. 8a–c). Recently, RIF1 was reported to promote compaction of DSB-flanking chromatin[53]. Consistently, depletion of 53BP1, RIF1, or ASF1 impaired chromatin condensation at the LacO array after induction of DSBs (Fig. 5b, c and Supplementary

Fig. 8d), demonstrating that 53BP1, RIF1, and ASF1 promote changes in high-order chromatin structure flanking DSBs. ATM-dependent phosphorylation of 53BP1 mediates its interaction with RIF1[4]. Therefore, ATM condenses chromatin flanking DSBs, possibly by regulating the 53BP1-RIF1 pathway.

Both RIF1[54–59] and ASF1[24–26] promote heterochromatin assembly in multiple species. Given that RIF1–ASF1 accumulates H3K9me1 (Fig. 1b), a precursor of H3K9me3 for heterochromatin, we thus assessed the impact of RIF1–ASF1 on heterochromatinization at the LacO array upon DNA damage. Because HP1γ stimulates NHEJ, while HP1α and β promote HR[58,60], we focused on heterochromatin marks HP1γ and H3K9me3. The signals of HP1γ and H3K9me3 were significantly increased at the array after DSB induction (Supplementary Fig. 8a–c), indicating the formation of heterochromatin or heterochromatin-like structures. Interestingly, these signals were decreased when 53BP1, RIF1 or ASF1 was depleted (Fig. 5b, c). The decreased accumulation of H3K9me3 after depletion of 53BP1, RIF1 or ASF1 was further confirmed by chromatin immunoprecipitation (ChIP; Fig. 5d and Supplementary Fig. 8e). These results demonstrate that 53BP1, RIF1, and ASF1 promote chromatin condensation after DNA damage through heterochromatinization.

We examined whether the interaction of RIF1 with ASF1 is required for this function. Wild-type RIF1 and ASF1a, but not their interaction-defective mutants, promoted chromatin condensation and H3K9me3 accumulation flanking DSBs in RIF1- and ASF1a/b-depleted cells, respectively (Fig. 5e–h and Supplementary Fig. 8f, g). Moreover, ASF1a V94R lost this function (Fig. 5e, f and Supplementary Fig. 8f). These results demonstrate that RIF1 and ASF1 promote heterochromatinization at DSB sites as a complex through its histone chaperone activity.

Heterochromatin marks have both positive and negative functions in HR[60–65], supposedly occurring in a temporally separated manner[66]. Chromatin condensation surrounding DSBs via tethering of HP1 or KAP1 suppresses BRCA1-mediated resection[60]. We determined whether 53BP1-, RIF1-, or ASF1-mediated chromatin condensation at the array impacts resection. Depletion of 53BP1, RIF1 or ASF1 significantly increased the signals of BRCA1 and RPA32 at the array after DSB induction in the G1 phase (Supplementary Fig. 9a–d), which is consistent with our finding that 53BP1-, RIF1-, or ASF1-dependent chromatin condensation antagonizes BRCA1-dependent end resection.

We determined whether 53BP1-, RIF1- or ASF1-mediated heterochromatinization flanking DSB sites is limited to the artificial sites of the LacO array. All of the GFP-tagged HP1 paralogs (α, β, and γ) were recruited to laser-induced DSB sites in a 53BP1-, RIF1- and ASF1-dependent manner (Supplementary Fig. 10a–f). Moreover, similar to 53BP1, RIF1, and ASF1, GFP-HP1γ distributed to chromatin distal to DSBs, and this distribution was significantly decreased when 53BP1, RIF1, or ASF1 was absent (Supplementary Fig. 10g). These results suggest that 53BP1-RIF1–ASF1-mediated heterochromatinization may occur at common DSB sites.

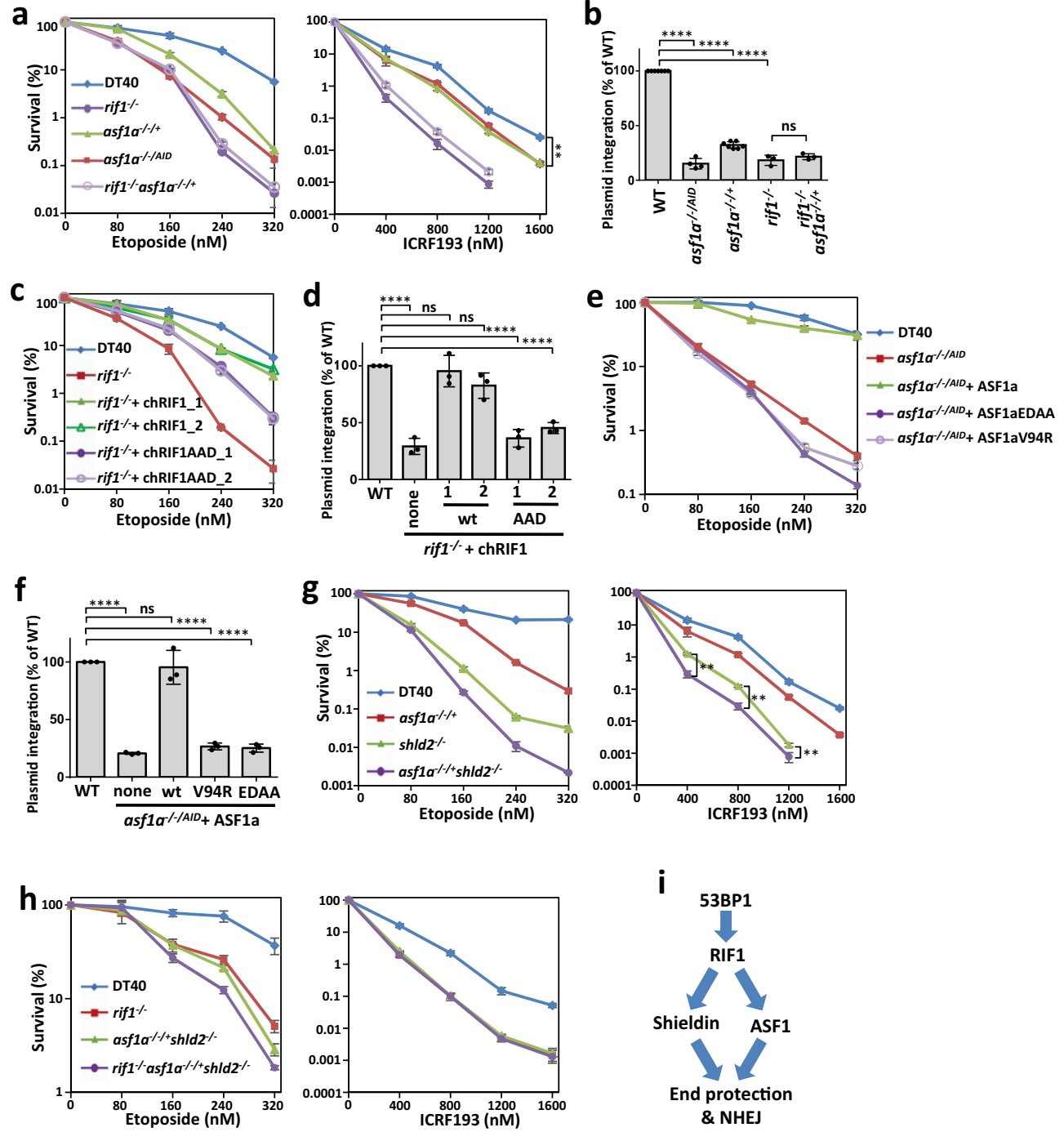

**Fig. 3 ASF1 promotes NHEJ through its interaction with RIF1 and histone chaperone activity. a**, **c**, **e**, **g**, **h** Etoposide- or ICRF193-sensitivity assays of various groups of DT40 cells. The mean and s.d. from three independent experiments are shown. **b**, **d**, **f** Random integration assays of various groups of DT40 cells. Data are expressed as the mean number of puromycin-resistant colonies ± s.d.; $n = 7, 4, 7, 3$, and 3 replicates from left to right in (**b**); $n = 3$ replicates in (**d**, **f**). ns, $P > 0.05$; **$P < 0.01$; ****$P < 0.0001$. Statistical analysis was performed using two-way $t$ test (**a**, **g**) or one-way ANOVA (**b**, **d**, **f**). Statistical source data, including the precise $P$ values, are provided in the source data. **i** Cartoon showing the genetic relationships of 53BP1, RIF1, Shieldin, and ASF1.

HP1 was reported to be recruited to laser-induced DSBs by CAF-1 p150[67]. We examined whether ASF1 recruits HP1 through CAF-1. No significant change in GFP-HP1γ recruitment to laser-induced DSBs was detected after depleting CAF-1 p60 or p150 in wild type or $ASF1a^{\Delta}ASF1b^{H}$ cells (Supplementary Fig. 11a–f), suggesting that CAF-1 is not important for HP1 recruitment in our system. This discrepancy may be due to differences between laser-induction systems or cell lines used in different studies.

**SUV39h1/2 acts downstream of ASF1 to promote heterochromatinization at DSB sites and antagonize BRCA1-dependent HR.** SUV39H enzymes catalyze the conversion of H3K9me1 to H3K9me3[68] and promote transient heterochromatinization at DSB sites[69]. Consistently, SUV39h1 was relocalized to laser-induced DNA damage sites, similar to 53BP1, RIF1, and ASF1 (Fig. 6a, b). SUV39h1 recruitment was dramatically decreased in 53BP1-, RIF1-, or ASF1a/b-deficient cells (Fig. 6a, b), suggesting that SUV39H enzymes may play a role in

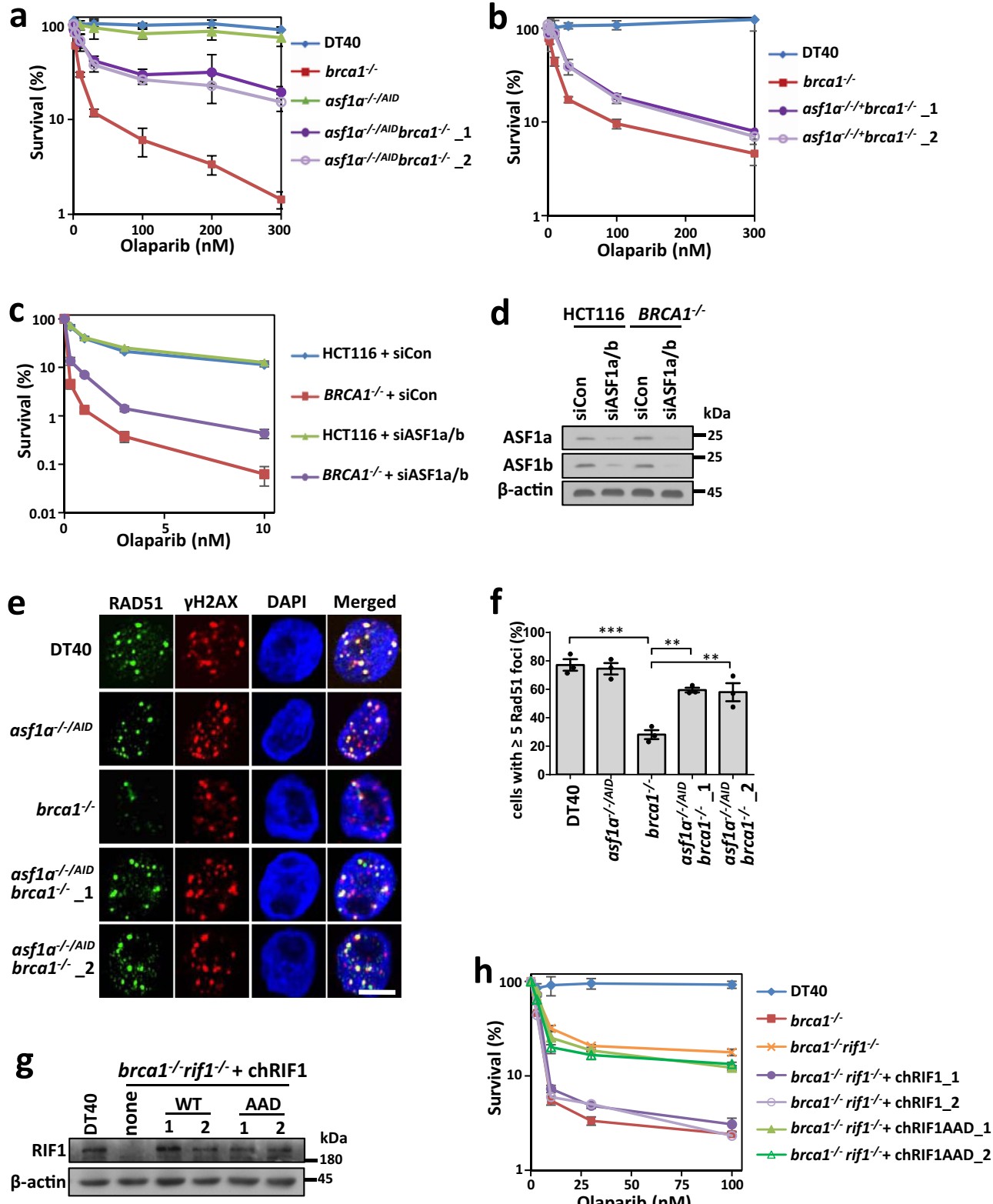

DSB repair downstream of 53BP1-RIF1–ASF1. The recruitment of SUV39H may be mediated by ASF1-provided H3K9me1 through the affinity between the enzyme and its substrate, and/or by its interaction with RIF1 as previously described[59].

Depletion of SUV39h1 and SUV39h2 prevented chromatin condensation, reduced the accumulation of heterochromatin marks, and promoted the recruitment of BRCA1 and RPA32 to the array

after DSB induction in U2OS-265 cells (Fig. 6c–h). It has been reported that HP1α accumulates at laser-induced DNA damage sites in both wild-type and SUV39h1/2 double-knockout MEF cells[67]. We detected obvious accumulation of GFP-HP1γ at laser-induced DSBs in both wild-type and SUV39h1/2 double-knockout HEK293T cells, but the signal in double-knockout cells was modestly decreased (Supplementary Fig. 12a–c), in agreement with

**Fig. 4 ASF1 suppresses end resection and HR in BRCA1-deficient cells. a, b** PARPi (olaparib) sensitivity assay of $brca1^{-/-}$, $asf1a^{-/-/AID}$, $asf1a^{-/-/AID}brca1^{-/-}$ (**a**) and $asf1a^{-/-/+}brca1^{-/-}$ (**b**) DT40 cells. The mean and s.d. from three independent experiments are shown. **c** PARPi sensitivity assay for ASF1- and BRCA1-deficient HCT116 cells. The mean and s.d. of the results from three independent experiments are shown. **d** Immunoblots showing the knockdown efficiency of BRCA1 in HEK293T cells. **e, f** Immunofluorescence (**e**) and quantification (**f**) showing that ASF1a blocks end resection in BRCA1-deficient DT40 cells. Cells were treated with 4 Gy X-ray radiation and released 4 h before fixing. The mean and s.d. from three independent experiments are shown. Scale bar, 5 μm. **P < 0.01; ***P < 0.001. Statistical analysis was performed using one-way ANOVA. **g** Immunoblots showing protein levels of RIF1 after complementation. **h** Olaparib-sensitivity assay of the $brca1^{-/-}rif1^{-/-}$ DT40 cells complemented with wild-type or mutated chicken RIF1. The mean and s.d. from three independent experiments are shown. Uncropped images of the gels and statistical source data including the precise P values are provided in Source data.

another study that recruitment of SUV39h1, KAP1, and HP1 at DSBs is interdependent, possibly leading to cycles of KAP1-HP1-SUV39h1 loading and spreading of H3K9me3 along the chromatin[69]. These results are consistent with our hypothesis that SUV39h1/2 promotes heterochromatinization surrounding DSBs and suppresses BRCA1-dependent end resection.

Importantly, disruption of SUV39h1/2 leads to reduced cellular resistance to etoposide and random integration of foreign DNA (Fig. 6i, j and Supplementary Fig. 13a–e), suggesting that SUV39h1/2 promotes NHEJ. Moreover, the absence of SUV39h1/2 rescued PARPi sensitivity of $brca1^{-/-}$ DT40 cells (Fig. 6k), demonstrating that SUV39H enzymes play a role in antagonizing BRCA1 to suppress HR. The rescue effect of the absence of SUV39h1/2 was weaker than that of deficiency in 53BP1 or ASF1a (Fig. 6k). This difference may have been due to two reasons: one is the activity of a potential parallel pathway to SUV39h1/2 for H3K9 trimethylation; the other is neutralization via the loss of other functions of SUV39h1/2, such as promotion of ATM activity[67,69,70], which is required for PARPi resistance in 53BP1/BRCA1 double-knockout cells[1]. These findings are in agreement with our conclusion that ASF1-SUV39H1/2-axis-dependent heterochromatinization antagonizes BRCA1 to suppress HR and promote NHEJ.

## Discussion

**RIF1–ASF1 promotes formation of high-order chromatin structure to protect broken ends.** It has been long proposed that the 53BP1-RIF1 pathway antagonizes BRCA1-dependent end resection possibly by changing high-order chromatin structure and subsequently controlling nuclease access[16,22,71]. Here, we identified ASF1 as a partner of RIF1 to protect broken ends through heterochromatinization of the DSB-flanking region in chicken DT40 cells (Fig. 7). It is well established that heterochromatin and heterochromatin-like structures represent relatively condensed chromatin that is inaccessible for nucleases. Limiting the accessibility of chromatin neighboring DNA ends not only suppresses excessive resection but also inhibits its initiation (Fig. 7). This function is quite different from that of shieldin-CST-Polα, which suppresses resection after initiation. Consistently, our data reveal that ASF1 and the shieldin complex act at two parallel pathways to repair DSBs.

Although both RIF1 and ASF1 are conserved in flies and yeast, no obvious B-domain presents in fly or yeast RIF1, implying that the RIF1–ASF1 complex may only exist in vertebrates. Assembly of the RIF1–ASF1 complex is dynamically regulated and significantly enhanced in response to DNA damage, although it mimics that of histone chaperone complexes CAF-1-ASF1 and HIRA-ASF1. Moreover, upon DSBs, the RIF1–ASF1 complex accumulates H3K9me1, a precursor of H3K9me3, on nucleosomes. It remains to be investigated whether RIF1, like CAF-1 and HIRA, has direct histone chaperone activity in the process of H3–H4 deposition using in vitro biochemical assays, or whether it is only a platform that provides ASF1-H3–H4 to other histone chaperones. Chromatin around DSBs transiently expands (reaching a maximum at ~1.5 min), followed by hypercondensation

beyond the predamage baseline level at 20–30 min[70]. If the expansion is accompanied by nucleosome disassembly, RIF1–ASF1 may promote nucleosome re-assembly during subsequent condensation. Otherwise, RIF1–ASF1 may directly promote nucleosome exchange activity to deposit H3K9me1, which is converted to H3K9me3 for the formation of heterochromatin by SUV39H1/2[68]. Recently, super-resolution imaging revealed that 53BP1 and RIF1 play a shieldin-independent role in stabilization of compact chromatin topology via an unknown mechanism[53]. The question of whether ASF1 acts together with 53BP1 and RIF1 in safeguarding chromatin topology around DSBs remains open for future study.

**ASF1 plays multiple functions in DSB repair.** It was reported that ASF1 promotes MMS22L-TONSL-mediated RAD51 loading onto ssDNA during HR[44]. Consistently, we detected localization of ASF1 on the ssDNA-region in the HIRDC assay (Fig. 2f, g), and ASF1-defcient cells also showed mild sensitivity to CPT at low dosages (Supplementary Fig. 6f). It has been proposed that NHEJ makes the first attempt to repair DSBs and, if rapid rejoining is not achieved, then the DNA ends are resected and repaired via HR[72]. The functions of ASF1 in HR and NHEJ may be temporally and spatially separated: ASF1 is initially recruited to the broken ends by 53BP1-RIF1 for NHEJ; if NHEJ repair does not ensue, then after end resection, accumulated ASF1 around DSBs may in turn promote RAD51 loading for HR at ssDNA regions and simultaneously prevent over-resection at adjacent chromatin regions, respectively.

Moreover, ASF1a was reported to interact with MDC1 and promote NHEJ in a manner independent of its histone chaperone activity[73]. Indeed, unlike RIF1 and MMS22L-TONSL, MDC1 was not detected by mass spectrometry analysis in ASF1a immunoprecipitates from both our (Fig.1a) and other groups[34–36], indicating that the interaction between ASF1a and MDC1 may be quite weak or transient. Histone chaperone activity of ASF1a was reported to be dispensable for MDC1-mediated signaling in human U2OS and HeLa cells, and histone-binding-defective ASF1a (V94R) promoted NHEJ as effectively as the wild-type form in NHEJ/DsRed293B cells, which contain a NHEJ reporter system[73]. However, in this study, V94R could not recover the NHEJ activity of $asf1a^{-/-/AID}$ chicken DT40 cells in the etoposide-sensitivity analysis or random integration assay (Fig. 3e, f). These discrepancies may be due to differences in assays and/or cell lines. In other words, the importance of the histone chaperone activity-dependent and -independent functions of ASF1a in NHEJ may depend on species, cell type and/or microenvironment. The manner in which the two functions of ASF1a in NHEJ are coordinated in different conditions remains to be investigated in the future.

**Heterochromatin impacts both the efficiency and the balance of HR- and NHEJ-mediated repair of DSBs.** Heterochromatic DSBs are repaired more slowly than euchromatic lesions, reflecting that the compacted structure of heterochromatin impairs the overall efficiency of DSB repair, possibly due to limited accessibility of the sites for repair factors[74,75]. On the

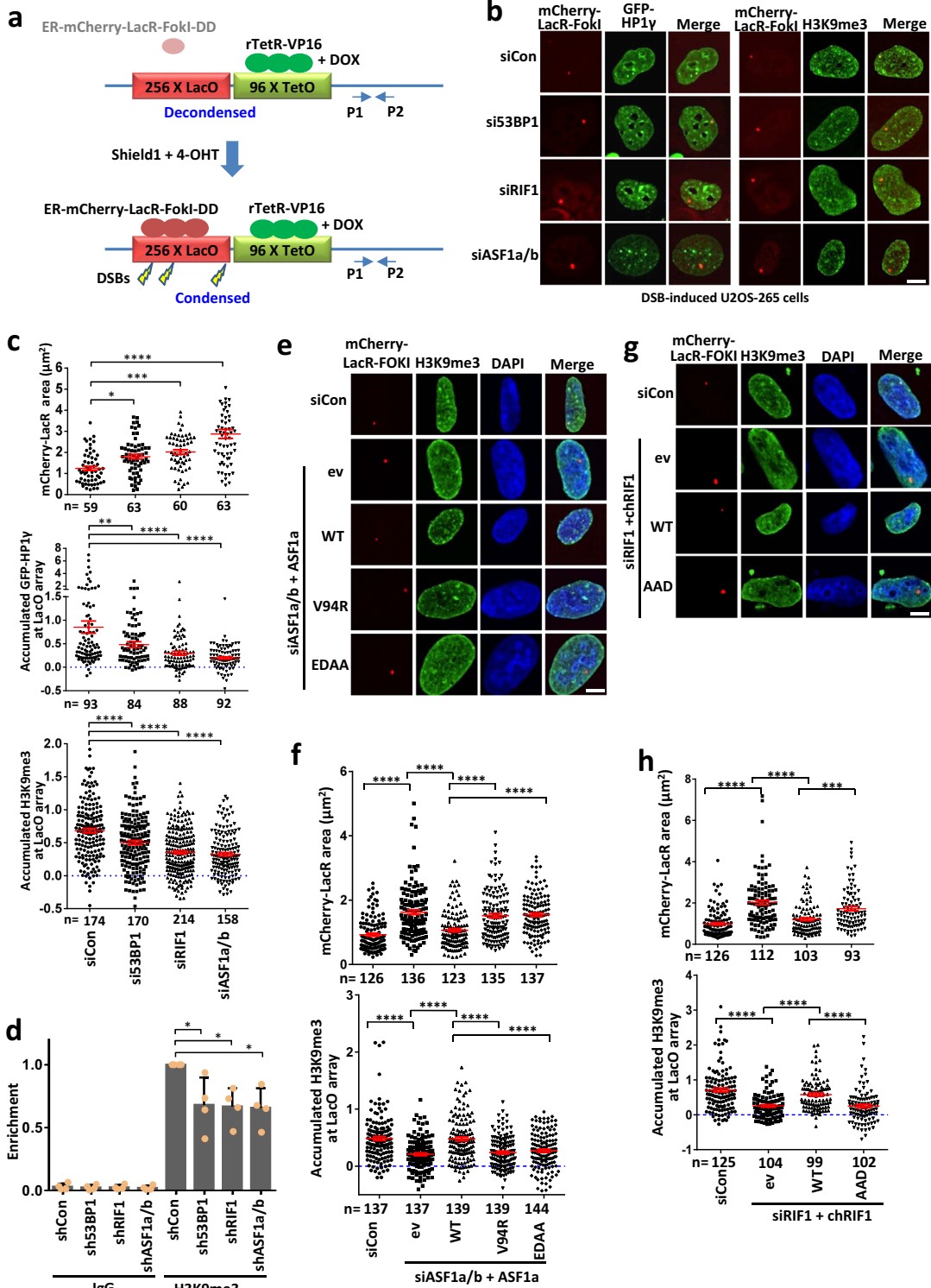

other hand, the balance between HR and NHEJ is strongly influenced by chromatin structure. DSBs that occur in the more open, active chromatin environment of euchromatin are readily repaired by HR, while those occurring in closed, repressive heterochromatin are generally repaired by NHEJ[60,76,77]. This mechanism may have evolved to avoid genomic instability mediated by HR repair at repetitive DNA elements, which are abundant at heterochromatin[66,78]. Therefore, the pre-existing heterochromatin status may facilitate the formation of the high-order structure by 53BP1-RIF1–ASF1 after DNA damage, tilting

the balance towards NHEJ repair. Therefore, these studies are in agreement with our model.

Several lines of evidence suggest that heterochromatin marks (HP1, KAP1, SUV39H, H3K9me2, H3K9me3, and macroH2A) and chromatin condensation could take place surrounding DSBs. These marks have both positive[62–65] and negative[60–62] roles in BRCA1-mediated resection and HR. Moreover, heterochromatinization also promotes ATM-dependent DNA damage response signaling[67,69,70]. These contradictory findings can be reconciled by a model in which chromatin condensation may play multiple

**Fig. 5 53BP1-RIF1–ASF1 condenses chromatin flanking DSBs. a** Scheme of U2OS-265 DSB reporter cells. Doxycycline (DOX) induces binding of rTetR-VP16 to chromatin and subsequent chromatin decondensation. Shield1 and 4-OHT induce FokI-mediated DSBs within LacO repeats and subsequent chromatin condensation. The ChIP–quantitative PCR (qPCR) primers are labeled as p1–p2. **b, c** GFP-HP1γ and H3K9me3 signals (**b**) in the array and their quantification (**c**) in 53BP1-, RIF1- or ASF1a/b-depleted U2OS-265 cells after DSB induction. The mCherry-LacR area was also quantified in (**c**). The mean intensity in the nucleus was set as 0. Data are presented as mean values $+/-$ s.e.m. The numbers of cells pooled from two independent experiments are indicated. **d** ChIP-qPCR performed with an antibody to H3K9me3 in the U2OS-265 cells line with induction of DSBs after depletion of 53BP1, RIF1 or ASF1a/b. IgG was used as a control antibody. The mean and s.d. from 4 independent experiments are shown. **e–h** H3K9me3 staining (**e, g**) in the array and quantification (**f, h**) in ASF1a- (**e, f**) or RIF1-complemented (**g, h**) cells after DSB induction. The mean intensity in the nucleus was set as 0. Data are presented as mean values $+/-$ s.e.m. The numbers of cells pooled from three independent experiments are indicated. ev empty vector. Scale bar, 5 μm. $*P < 0.05$; $**P < 0.01$; $***P < 0.001$; $****P < 0.0001$. Statistical analysis was performed using one-way ANOVA. Statistical source data, including the precise $P$ values, are provided in the source data.

(even opposite) roles in DSB repair in a spatiotemporally dependent manner as described previously[66] (Supplementary Fig. 14). After a transient (~1.5 min) expansion, chromatin at DSB sites is condensed by heterochromatinization, and then re-relaxes[66,70,78]. The maintaining time for condensed chromatin is variable from less than 5 min to more than 30 min in different studies, possibly due to the use of different chromatin regions, cells, and methods[67,69,70,79,80]. In a systematic analysis of the kinetics of DNA repair proteins at laser-induced DNA damage sites, HP1 was found to be recruited to damage sites by at least two independent events with a removal halftime of more than 20 min and 60 min, respectively[79]. This heterochromatinization may be mediated by the 53BP1-RIF1–ASF1-SUV39H axis and protect the broken ends from resection for NHEJ repair in the early stage (Supplementary Fig. 14). When rapid repair by NHEJ does not ensue, the heterochromatin mark H3K9me3 then recruits TIP60 to DSBs to hyperacetylate H4[81,82], which leads to chromatin relaxation, promoting BRCA1-dependent end resection and HR[52]. TIP60 also acetylates and fully activates ATM[83], which results in KAP1 phosphorylation and releases the KAP1/HP1/SUV3-9 and CHD3 complexes, promoting the relaxation of heterochromatin[75,84]. Simultaneously, BRCA1 is recruited by HP1 to promote end resection and HR[63,65]. Therefore, the temporal and spatial coordination of chromatin dynamics plays a key role in the pathway choice to optimize DSB repair (Supplementary Fig. 14).

Suppression of histone demethylation in isocitrate dehydrogenase 1/2 (IDH1/2)-mutated human malignancies by accumulated 2-hydroxyglutarate causes hyper-trimethylation of H3K9, which confers exquisite sensitivity to PARPi[85,86]. Our data provide mechanistic insight into how H3K9me3-mediated chromatin compaction causes PARPi sensitivity. Overexpression of epigenetic enzymes such as SET and SETBD1, which cause chromatin compaction, is widely found in human cancers[87,88]. If our conclusions, which are mainly acquired from chicken DT40 cells, are applied to human tumor cells, future investigations should assess whether such epigenetic enzymes are biomarkers that could indicate the potential responses of cancers to PARPi therapy.

## Methods

**Cell culture and transfection**. HEK293T, U2OS, and NIH2/4 cells were cultured in DMEM medium supplemented with 10% fetal bovine serum (FBS; Invitrogen). HCT116 cells were cultured in RPMI 1640 medium with 10% FBS. HEK293 suspension cells were cultured in SMM 293-TI medium (Sino Biological Inc.) with 1% FBS and 1% glutamine in an incubator with shaking at 140 rpm. The cell lines were obtained from the ATCC and were not among those listed as commonly misidentified by the International Cell Line Authentication Committee. All cell lines were subjected to mycoplasma testing twice per month and found to be negative. The identity of the cell lines was validated by STR profiling (ATCC) and by analysis of chromosome number in metaphase spreads.

DT40 cells were cultured in RPMI 1640 medium with 10% FBS, 2% chicken serum, 10 mM HEPES and 1% penicillin-streptomycin mixture at 39.5 °C (5% CO$_2$). Transfection was performed by electroporation using a Lonza Nucleofector 4D instrument. For selection, growth medium containing G418 (2 mg mL$^{-1}$), puromycin (0.5 μg mL$^{-1}$), blasticidin (25 μg mL$^{-1}$), or histidinol (1 mg mL$^{-1}$) was used.

For knockdown, siRNAs targeting ASF1a (5′-AAGUGAAGAAUACGAUC AAGU-3′), ASF1b (5′-AACAACGAGUACCUCAACCCU-3′), RIF1 (5′-GCAGCU

UAUGACUACUAAA-3′), 53BP1 (5′-GAAGGACGGAGUACUAAUA-3′), SUV39 h1 (5′-ACCUCUUUGACCUGGACUA-3′), SUV39h2 (5′-UAAUUAUGCUU GU CAUUAGAG-3′), BRCA1 (5′-AAUGCCAAAGUAGCUAAUGUA-3′), CHD1 (5′-GCGGTTTATCAAGAGCTATAA-3′), CAF-1 p150 (5′- GGAGAAGGCGGAGA AGCAG-3′), and CAF-1 p60 (5′-AAUCUUGCUCGUCAUACCA-3′) were trans fected using RNAi MAX (Invitrogen). To produce the 53BP1 (CCGGGATACTT GGTCTTACTGGTTTCTCGAGAAACCAGTAAGACCAAGTATCTTTTTTG), R IF1 (CCGGGCTCTATTGTTAGGTCCCATTCTCGAGAATGGGACCTAACAA TAGAGCTTTTTTG), ASF1a (CCGGGCACCTAATCCAGGACTCATTCTCG AGAATGAGTCCTGGATTAGGTGCTTTTTTG) and ASF1b (CCGGGAGTGGAA GATCATTTATGTTCTCGAGAACATAAATGATCTTCCACTCTTTTTG) shR NA, lentiviral plasmids were co-transfected into 293 T cells using PEI. After 4 days, the supernatants containing the packaged lentivirus were harvested and stored at −80 °C until further use.

**Antibodies**. Antibodies are listed below:

| Antibody designation | Source or reference | Identifiers | Additional information |
|---|---|---|---|
| Asf1a (rabbit polyclonal) | Proteintech (China) | 22259-1-AP | WB:1:2000 IF: 1:200 |
| Asf1b (rabbit polyclonal) | Proteintech (China) | 22258-1-AP | WB:1:2000 |
| Histone3 (rabbit polyclonal) | Novus Biologicals (Littleton, USA) | NB500-171 | WB:1:2000 |
| RIF1 (rabbit polyclonal) | homemade | | WB:1:2000 |
| 53BP1 (rabbit polyclonal) | Abcam (Cambridge, UK) | ab36823 | WB:1:2000 |
| 53BP1 (rabbit polyclonal) | Novus Biologicals (Littleton, USA) | NB100-304 | IF:1:250 |
| BRCA1 (rabbit polyclonal) | Millipore (St. Louis, MO, USA) | 07-434 | WB: 1:5000; IF:1:1000 |
| BRCA1 (mouse monoclonal) | Santa Cruz (Dallas, TX, USA) | sc-6954 | WB:1:100; IF:1:40 |
| RAD51 (rabbit polyclonal) | Santa Cruz (Dallas, TX, USA) | sc-8349 | IF:1:250 |
| RAD51 (rabbit polyclonal) | Abcam (Cambridge, UK) | ab133534 | IF:1:250 |
| γH2AX (mouse monoclonal) | Millipore (St. Louis, MO, USA) | 05–636 | IF:1:5000 |
| RPA32 (rabbit polyclonal) | Bethyl (Montgomery,TX, USA) | A300-244A | IF:1:250 |
| Histone H3K9me3 (rabbit polyclonal) | Abcam (Cambridge, UK) | ab176916 | IF:1:1000 |
| Histone H3K9me3 (rabbit polyclonal) | Abcam (Cambridge, UK) | ab8898 | IF:1:250 |
| Brdu (mouse monoclonal) | Becton Dickinson | 347580 | IF:1:50 |
| MRE11 | Abcam (Cambridge, UK) | ab214 | IF:1:250 |
| SUV39H1 (mouse monoclonal) | Millipore (St. Louis, MO, USA) | 05-615 | WB:1:1000 IF: 1:250 |
| SUV39H2 (rabbit polyclonal) | Abcam (Cambridge, UK) | ab190870 | WB:1:1000 |
| β-actin (mouse monoclonal) | MBL (Japan) | M177-3 | WB:1:1000 |
| anti-rabbit IgG Alexa Fluor 594 secondary antibodies | Invitrogen | A21207 | IF:1:250 |
| anti-mouse IgG Alexa Fluor 594 secondary antibodies | Invitrogen | A21203 | IF:1:250 |
| anti-rabbit IgG Alexa Fluor 488 secondary antibodies | Jackson Immunoresearch | 711-546-152 | IF:1:250 |
| anti-mouse IgG Alexa Fluor 488 secondary antibodies | Invitrogen | A21202 | IF:1:250 |

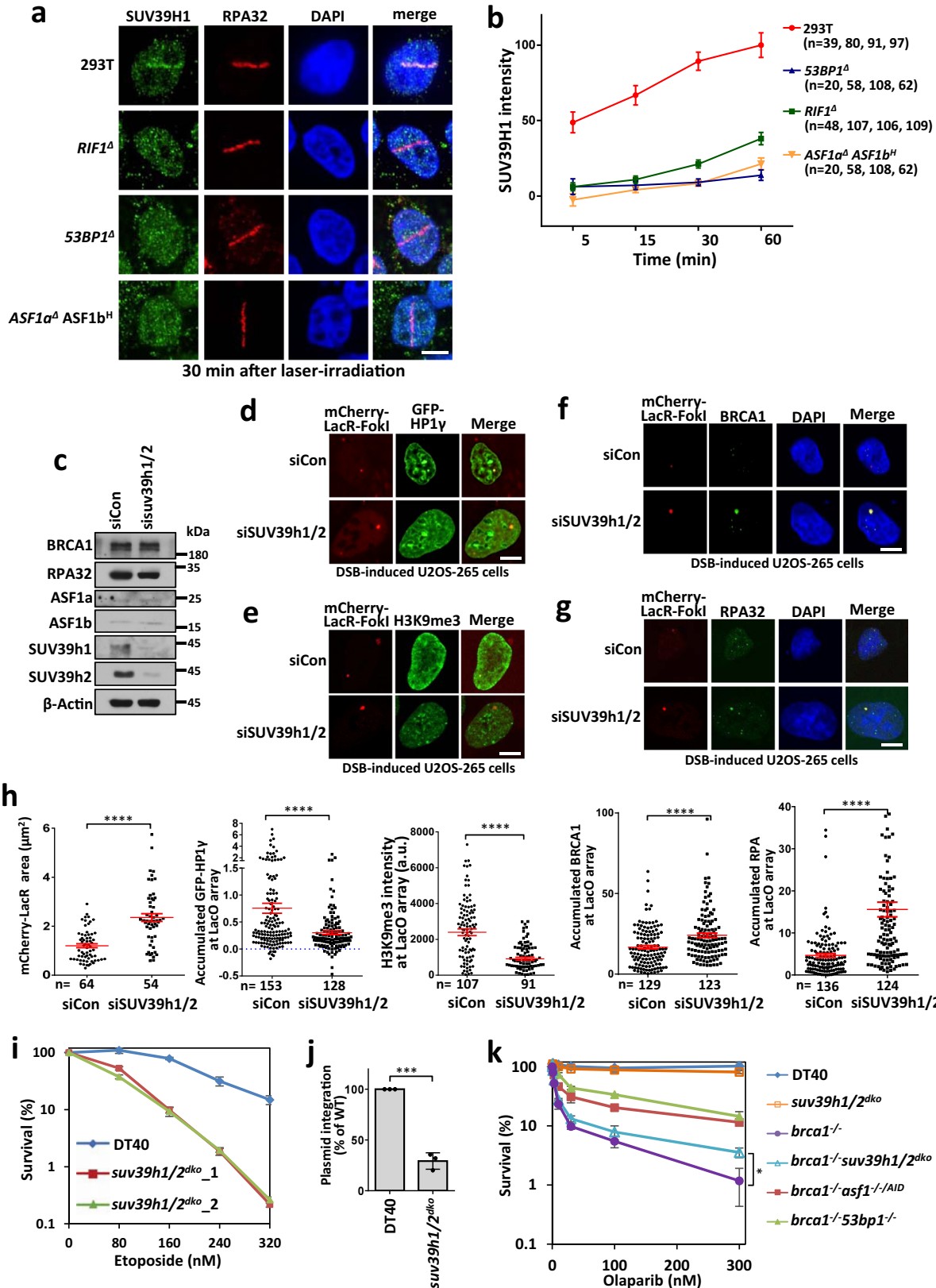

**Immunoprecipitation/MBP pulldown**. For immunoprecipitation (IP), expression plasmids were transfected into HEK293 suspension cells with polyethyleneimine. HEK293 cells were harvested 64 h after transfection, after which the pellet was lysed with NTEN buffer (20 mM Tris-HCl [pH 7.5], 150 mM NaCl, 10% glycerol, 0.5% NP-40, 10 mM NaF, 1 mM phenylmethylsulfonyl fluoride (PMSF), 1 μg mL$^{-1}$ leupeptin, and 1 μg mL$^{-1}$ aprotinin). The lysate was subjected to

ultracentrifugation at 440,000×$g$ for 15 min, after which the supernatant incubated with anti-FLAG M2 conjugated agarose beads at 4 °C for 4 h. The beads were washed four times with IP buffer (20 mM Tris-HCl [pH 7.5], 150 mM NaCl, 5 mM MgCl$_2$, 10% glycerol, 0.1% NP-40, 1 mM DTT, and 1 mM PMSF) and incubated with 400 μg mL$^{-1}$ 3× Flag peptide in IP buffer for 1–2 h. Subsequently, the eluted complexes were analyzed with sodium dodecyl sulfate-polyacrylamide gel

**Fig. 6 SUV39h1/2 compacts chromatin and antagonizes BRCA1-dependent resection to promote NHEJ. a, b** Recruitment of SUV39h1 (**a**) to laser-induced DNA damage sites in wild-type, 53BP1$^Δ$, RIF1$^Δ$ or ASF1a$^Δ$ASF1b$^H$ HEK293T cells and quantification (**b**). The mean and s.e.m. values are shown for every time point. **c** Immunoblots showing the knockdown efficiency of SUV39h1/2 in U2OS-265 cells. **d–h** GFP-HP1γ (**d**), H3K9me3 (**e**), BRCA1 (**f**), and RPA32 (**g**) signals in the array and quantification (**h**) in SUV39h1/2-depleted U2OS-265 cells after DSB induction. The mCherry-LacR area was also quantified in (**h**). For BRCA1 and RPA32 staining, cells were synchronized into the G1 phase as described previously[91]. Some experiments were carried out together with those in Fig. 5b, c and thus the same controls were used. The error bars represent the s.e.m. The numbers of cells pooled from two independent experiments are indicated. **i, j** Etoposide-sensitivity assay (**i**) and random integration assay (**j**) of wild-type and SUV39h1/2 double-knockout DT40 cells. The mean and s.d. of the results from three independent experiments are shown. **k** Olaparib-sensitivity assay of brca1$^{−/−}$, suv39h1/2$^{dko}$ and suv39h1/2$^{dko}$brca1$^{−/−}$ DT40 cells. The mean and s.d. of the results from three independent experiments are shown. *$P < 0.05$; ***$P < 0.001$; ****$P < 0.0001$. Statistical analysis was performed using the two-tailed $t$ test. Uncropped images of the gels and statistical source data including the precise $P$ values are provided in Source data.

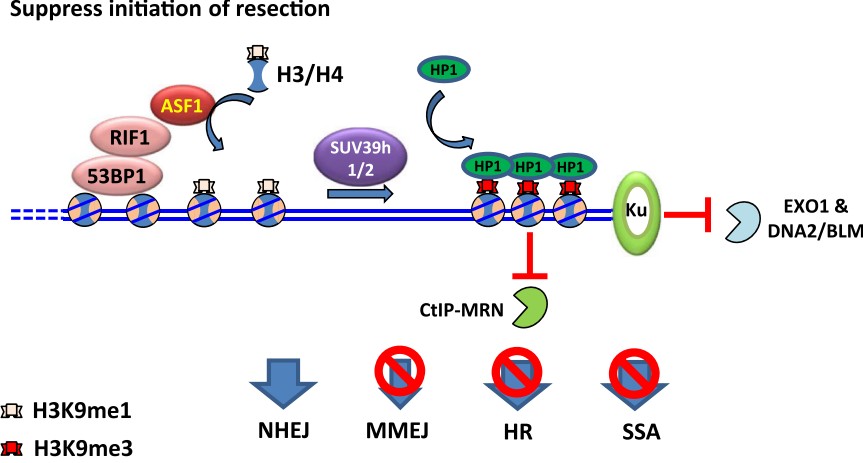

**Fig. 7 Model of suppression of BRCA1-dependent resection by the RIF1–ASF1 complex.** DSB ends are recognized and bound by Ku, which prevents directly resection by EXO1 or DNA2/BLM. 53BP1 binds to adjacent chromatin and recruits the RIF1–ASF1 complex, which provides H3–H4 heterodimers with K9me1 on H3 for nucleosome assembly/exchange. SUV39h1/2 then converts H3K9me1 to H3K9me3, which recruits HP1, leading to heterochromatinization. Chromatin condensation by heterochromatinization suppresses CtIP-MRN-mediated initiation of resection, and inhibits microhomology-mediated end joining (MMEJ), HR, and SSA. When NHEJ repair does not ensue, 53BP1 and the RIF1–ASF1 complex will be excluded from the chromatin proximal to the breaks by BRCA1 for initiating resection, but will retain at the distal chromatin to restrict resection and prevent SSA.

electrophoresis (SDS-PAGE). Chromatin fractions were prepared as described previously[89].

For MBP pulldown, 30 μg pDEST26–MBP–RIF1_B or pDEST26–MBP was transfected into a 30-mL suspension of HEK293 cells using polyethyleneimine. After 64 h, cells were harvested and lysed in 3 mL NTEN buffer. After ultracentrifugation at 440,000×$g$ for 15 min at 4 °C, the supernatant was incubated with amylose resin at 4 °C for 4 h. The beads were washed four times with NTEN buffer and eluted with 50 μL of phosphate-buffered saline (PBS) containing 20 mg mL$^{−1}$ of maltose.

**Mass spectrometry analysis of immunoprecipitated protein complexes.** The immunoprecipitated complexes were precipitated by trichloroacetic acid. Each sample was dissolved with 20 μl 8 M urea, and was successively incubated with 5 mM DTT and 10 mM IAA at 37 °C for 1 h. Then the sample was added with 75 μl 25 mM NH$_4$HCO$_3$ containing trypsin and incubated at 37 °C overnight. The reaction was stopped by adding 4 μl 2.5% TFA, and the peptide segments were desalted and dried in a vacuum concentrator.

Peptides were solubilized in 0.1% formic acid, and were separated by an Easy nLC 1200 system (Thermo Scientific, USA), which contains a trap column (Acclaim PepMapTM100 75 μm × 2 cm nanoViper, C18, 3 μm,100 Å, Thermo) and an analytical column (Acclaim PepMapTM RSLC 75 μm × 15 cm nanoViper, C18, 2 μm,100 Å, Thermo), with the following HPLC gradient: 0.1% formic acid in water with 4–8% eluting buffer (0.1% formic acid in 80% acetonitrile) in 7 min, with 8–20% eluting buffer in 68 min, with 20–30% eluting buffer in 15 min, with 30–90% eluting buffer in 13 min and with 90% eluting buffer for 2 min at a flow rate of 280 nL/min. The eluted peptides were sprayed into a Thermo Fusion Lumos mass spectrometer (Thermo Scientific, USA) equipped with a nanospray source. The mass spectrometer was operated in data-dependent mode with an Orbitrap MS1 scan (300–1500 $m/z$, AGC target = 4 × 10$^5$) at a resolution of 60000, followed by HCD MS2 spectra on the 10 most abundant precursor ions detected by Orbitrap scanning (110 M/z, AGC target = 1 × 10$^5$) at a resolution of 15,000. For dynamic exclusion, isolation window was 1.6 m/z. Raw data were directly analyzed against the human proteome sequence database (Uniprot) restricted to Homo sapiens using Proteome Discoverer 2.2 (Thermo Fisher Scientific). The search parameters

were as follows: (a) trypsin as an enzyme with up to two missed cleavages; (b) carbamidomethylation of cysteine as a fixed modification; (c) oxidation of methionine and acetylation of protein N terminus as variable modifications; (d) precursor mass tolerance of 10 ppm; (e) fragment mass tolerance of 0.02 Da; and (f) Minimum peptide length of 6 aa. Peptides and proteins were filtered at a false discovery rate (FDR) of 1%.

Immunoprecipitation followed with mass spectrometry of FLAG_RIF1, FLAG_Asf1a, and FLAG_Asf1b was performed in two independent experiments, while that of the control sample was performed once.

**Generation of human knockout cells.** HEK293T knockout cells were generated using the CRISPR/Cas9 genome-editing system. Briefly, pX330 plasmids[90] containing the guide sequences (TACAAACATATGCCTTCCTG for ASF1a, GATG AACTCCTGTCCATGGT for ASF1b, ACCTCTGACCAGAGAGCTGC for 53BP1, CTCCCCCATACCCTACAAGT and CCCCATACCCTACAAGTCGG for RIF1 and GAAGGTAAAGAACCTGCAAC for BRCA1) were transfected into cells. Single colonies were picked after 10–14 days of culture. The isolated single colonies were subjected to western blotting and DNA sequencing to verify protein loss.

RIF1 knockout HEK293T cells and BRCA1 knockout HCT116 cells were generated as described previously[13,46].

**Generation of the DT40 knockout strains.** The MultiSite Gateway Three-Fragment Vector Construction Kit was used to generate DT40 knockout constructs for the asf1a gene. The primer pairs GGAGCTGTGTATCAGGTTGGTATGTTA G/CCCAGCACCTGAGTAGAGACTCTATG and CCTGAAGAGCAAAGCTCT TATTAGAGGAAG/CAGCCACATCACCCATACCATGAAC were used to amplify the 5′ and 3′ arms from genomic DNA, respectively. The 5′ and 3′ arms were cloned into the pDONR P4-P1R and pDONR P2R-P3 vector, respectively. A knockout construct was generated by attL × attR recombination of the pDONR-5′ arm, pDONR-3′ arm, pDONR-211 resistance gene cassette and pDEST R4-R3 target vector. The C-terminal region of the asf1a gene was amplified from genomic DNA using the primer pair CGGGGTACCATGGCAAAGGTTCAGGTGAA/

CCGGAATTCTCACATGCAGTCCATGTGGG and cloned into the pAID1.1-C vector. The knockout constructs were linearized before transfection. The $asf1a^{-/-/AID}$ cells were verified by western blotting.

To generate knockout constructs of SUV39h1 and SUV39h2, the 5′ arm and 3′ arm were amplified from chicken genomic DNA using the primers AAGCACA GAGTGGTTGGGTT/AACCCACTTCGGGAGCATTTG (for the SUV39h1 3′ arm), TCCCTCCACCCGCAATAAAC/GTAACCCACTTCCGAGGGGTG (for the SUV39h1 5′ arm), GTATGTATTTATCTCATGTGGATTATTTTGAAGGACAA AACAGGAG/CTGGATGAGCTCAGACCATCAGCAGAG (for the SUV39h2 3′ arm) and AGGGCTGGCGAGGAGGTGAG/CTTTAAATACAGAAACAGATA ACAGATTTAGCCATGGTTTCAATTC (for the SUV39h2 5′ arm). The first arm was cloned into the pClone007 vector (Beijing Tsingke biotech TSV-007S) using the pClone007 simple vector kit. The second arm and the resistance genes were successively inserted into the pClone007-3′ arm. The knockout construct was linearized before transfection. The gene knockout clones were validated by genomic DNA PCR.

Generation of $rif1^{-/-}$, $shld2^{-/-}$, and $brca1^{-/-}$ DT40 cells was performed as described previously[4,13].

**Cell survival assay**. For the cell survival assay, 200–400 cells were plated into each well of a 96-well plate with a range of doses of olaparib. After 72 h of incubation, cells were pulsed for 4 h with CellTiter 96 Aqueous One solution reagent (Promega). Cell viability was measured by a luminometer, and each dosage was measured in triplicate. For camptothecin (CPT), a density of 1000–1500 cells per well was used, and the incubation period was 48 h.

To perform a clone formation assay using DT40 cells, 200–20,000 cells were seeded into each well of a six-well plate filled with 0.7% methylcellulose medium. The plates were treated with the appropriate dose of olaparib, etoposide or ICRF193. The number of colonies was counted after 7–14 days of incubation at 39.5 °C.

For the clone formation assay with HEK293T cells, 300–20,000 cells were seeded into each well of a six-well plate containing DMEM medium (10% FBS, 1% P/S). The plates were treated with the appropriate dose of olaparib or exposed to the appropriate dose of X-ray radiation. After 10 days of incubation at 37 °C, the number of colonies was counted.

**Random integration assay**. DT40 cells were transfected with a linearized pLox-puro plasmid. After 24 h, 100 cells were plated into 96-well plates to determine the cell plating rate, and one million cells were plated into 96-well plates with 0.5 μg mL$^{-1}$ puromycin for detection of the random integration efficiency. The number of clones was counted after 5–7 days of incubation. The random integration efficiency was normalized by the plating rate.

**Immunofluorescence**. Briefly, U2OS or 293T cells were seeded on polylysine-coated coverslips before the experiments. After washing with cold PBS, the cells were pre-extracted with 0.5% Triton X-100 in CSK buffer (20 mM HEPES [pH 7.0], 100 mM NaCl, 300 mM sucrose, and 3 mM MgCl$_2$) for 10 min at 4 °C. Next, the cells were washed three times with PBS and fixed with 3% paraformaldehyde for 10 min at room temperature. The cells were permeabilized for 10 min with 0.5% Triton X-100 in CSK buffer, washed three times with 0.05% Tween-20 in PBS and blocked with 5% BSA for 15 min. Next, the cells were incubated with the primary antibodies for 90 min. After washing, the cells were incubated with the secondary antibodies diluted in 1% BSA/PBS for 30 min. After three washes, the cells were mounted with ProLong Gold antifade reagent with DAPI (Invitrogen). Images were acquired with an ANDOR Dragonfly system on a Leica DMI8 microscope with a 100× oil immersion objective using Fusion.shell software.

**Laser-induced foci and HIRDC**. U2OS or HEK293 cells expressing GFP- or/and mCherry-fused proteins were cultured at 37 °C in DMEM medium containing 10% FBS. During microirradiation and imaging, the cells were maintained in a temperature-controlled container with 5% CO2 in glass-bottom dishes (NEST Biotechnology). The laser system (Micro-Point Laser Illumination and Ablation System, ANDOR; 365 nm wave) was directly coupled to a Leica DMI8 microscope with a ×100 oil immersion objective. Images were acquired with ANDOR IQ3 software through an ANDOR IXON camera with an ANDOR Dragonfly system.

In order to obtain images of unperturbed cells, time-lapse image acquisition was begun before laser microirradiation. Microirradiation was performed on an indicated line in the nuclei of cultured cells via micro-point laser illumination (65% output) at the time of the second image. Images were collected every 30 s for 10 min and analyzed with ImageJ software (NIH). Recruitment was measured by determining the mean fluorescence intensity within the damage region and normalizing this value to the mean fluorescent intensity of the unirradiated nucleus. For each cell, a separate region was measured for background subtraction. The relative fluorescence intensity was calculated by the following formula: $RFI(t) = (I_t - I_b)/(I_{nu} - I_b)$, where $I_t$ is the mean fluorescence intensity of the microirradiated region, $I_b$ is the mean fluorescence intensity of the background, and $I_{nu}$ is the mean fluorescence intensity of the unirradiated nucleus.

For the HIRDC assay, microirradiation was carried out within a dot (~1 μm in diameter) in the nucleus with a fill-in program. The micro-point laser illumination output was set at 65% unless indicated.

**FACS**. Cells were labeled with 10 μM BrdU (Sigma; B5002) for 20 min before harvest. After fixation with 70% ethanol, cells were resuspended in 2 ml 4 M HCl at room temperature for 20 min and then neutralized with 10 ml 0.1 M Na$_2$B$_4$O$_7$. Then cells were washed twice with 0.1% Triton-100 and 1% BSA in PBS, and were incubated with anti-BrdU antibody at room temperature for 30 min. After washing, the cells were incubated with anti-mouse IgG Alexa Fluor 488 secondary antibodies at room temperature for 30 min. After three times washing, cells were incubated with 50 μg/ml PI at room temperature for 30 min and finally analyzed with BD FACSVerse flow cytometer. The raw data were processed by FlowJo software.

**Imaging quantification in U2OS cells**. U2OS-265 cells expressing mCherry-LacR or GFP-rTetR were cultured at 37 °C in DMEM medium containing 10% FBS. Images were acquired with Imaris software (Bitplane) through an ANDOR IXON camera with an ANDOR Dragonfly system. The area of the array was analyzed using Imaris (Bitplane). Cells were chosen at random for the analysis of the intensity of the indicated signals. The region of the array was defined by the mCherry-LacR signal and measured using Imaris software. The accumulated signals were measured by determining the mean fluorescent intensity of the array region and normalizing this value to the mean fluorescent intensity of the entire nucleus. For each cell, a separate region was measured for background subtraction. The accumulated signal was calculated by the following formula: $I = (I_{array} - I_{nu})/(I_{nu} - I_b)$, where $I_{array}$ is the mean fluorescence intensity of the array region, $I_b$ is the mean fluorescence intensity of the background, and $I_{nu}$ is the mean fluorescence intensity of the nucleus. All intensity analysis was performed on unprocessed images using ImageJ software.

**ChIP assay**. Cells were cross-linked with 1% (v/v) formaldehyde for 10 min, followed by the addition of glycine to 0.125 M for 5 min to stop the cross-linking. Cells were lysed in lysis buffer (5 mM HEPES (pH 8.0), 85 mM KCl, 0.5% NP-40). Nuclei were isolated and resuspended in 50 mM Tris-HCl (pH 8.0), 10 mM EDTA (pH 8.0), 1% SDS, sonicated to obtain ~200–500 bp chromatin fragments using a Bioruptor (Diagenode). Chromatin fragments were precleared with salmon sperm DNA/protein-A sepharose slurry for 30 min-1 h at 4 °C with agitation. Then the chromatin fragments were diluted in 20 mM Tris-HCl (pH 8.0), 2 mM EDTA (pH 8.0), 150 mM NaCl, 1% Triton X-100, and incubated with rabbit IgG or anti-H3K9me3 antibodies on the rotating platform at 4 °C overnight. The salmon sperm DNA/protein-A sepharose beads slurry were added to each sample, incubate for 1–2 hr at 4 °C with agitation. Beads were washed once with low-salt buffer (20 mM Tris-HCl, pH 8.0, 2 mM EDTA, pH 8.0, 150 mM NaCl, 0.1% SDS, 1% Triton X-100), once with high salt buffer (20 mM Tris-HCl, pH 8.0, 2 mM EDTA, pH 8.0, 500 mM NaCl, 0.1% SDS, 1% Triton X-100), once with LiCl wash buffer (10 mM Tris-HCl, pH 8.0, 1 mM EDTA, pH 8.0, 250 mM LiCl, 1% NP-40, 1% deoxycholic acid) and twice with TE buffer (10 mM Tris-HCl, pH 8.0, 1 mM EDTA, pH 8.0). Then 100 μl of 10% (wt/vol) Chelex 100 slurry were added to the washed beads. The sample were boiled (100 °C) for 10 min and treated with Proteinase K at 55 °C for 30 min. The Proteinase was inactivated at 100 °C for 10 min. The DNA sample were transferred to new tubes after centrifugation. qPCR was carried out on a QuantStudio 6 Flex (Thermo Fisher) instrument using the SYBR Green detection system and primers GGCATTTCAGTCAGTTGCTCAA and TTGGCCGATTCATTAATGCA.

**RNA extraction and RT-qPCR**. RNA was extracted using TRIzol reagent (Invitrogen). mRNA were reversely transcribed with gDNA removal and cDNA synthesis kit (TransScript; AT311). qPCR was carried out on a QuantStudio 6 Flex (Thermo Fisher) instrument using the SYBR Green detection system (TransScript). Primers CTGGG CTACACTGAGCACC and AAGTGGTCGTTGAGGGCAATG for GAPDH, TCA AAACACAAAAGCCCAC and GAGTGTCCGTCTTTAAGGGTG for CAF-1-p60, AGTACCAGTCCCTTCCCC- and TCTTCCCTTTCTGCACGTAAC for CAF-1-p150 were used. The following amplification program was used: 94 °C for 30 s, 94 °C for 5 s and 60 °C for 30 s, 40 cycles.

**Statistics and reproducibility**. All immunoblots were performed at least three times unless otherwise noted in the legend. GraphPad Prism 6 and Excel 2013 were used for statistical analysis. Statistical significance was assessed using the two-tailed Student's $t$ test, one-way ANOVA, or two-way ANOVA. The data were normally distributed and the variance between groups being statistically compared were similar. No statistical methods or criteria were used to estimate sample size or to include/exclude samples. All image analyses were carried out in a double-blind approach. All the experiments in Fig. 1a–c, e; 2b-f; 4d, g; 6c and supplementary Figs. 1c; 2a-c, e-f, h; 3a-b; 4a, d; 5b; 6a, d, g, h; 7c; 8d-g; 10g; 12a; 13b, d, e. were repeated at least three independently with similar results.

**Reporting summary**. Further information on research design is available in the Nature Research Reporting Summary linked to this article.

## Data availability

The data that support this study are available from the corresponding author upon reasonable request. The mass spectrometry proteomics data have been provided in Supplementary Data 1 and deposited to the ProteomeXchange Consortium via the PRIDE partner repository with the dataset identifier PXD031311. Source data are provided with this paper.

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

## Acknowledgements

We thank Weidong Wang for his advice and revisions to the manuscript. We thank the Analytical Instrumentation Center in Peking University, the Imaging Core and Mass-spectrum Core at the National Center for Protein Sciences at Peking University for imaging and mass-spectrum analysis. U2OS-265 cells were a gift from Daniel Durocher under a permission of Roger A. Greenberg. This work was supported in part by Beijing Outstanding Young Scientist Program (BJJWZYJH01201910001005) to Q.L., the National Key Research and Development Program of China (2021YFA0909304) to D.X., and the National Natural Science Foundation of China (31870807 and 32071285 to R.G. and 82002991 to S.F.).

## Author contributions

S.F., S.M., K.L., S.G., S.N., J.S., R.G., Y.C., and B.B. carried out experiments. I.S., Q.L., R.G., and D.X. designed experiments and interpreted the results. S.F., S.M., K.L., and D.X. wrote the manuscript.

## Competing interests

The authors declare no competing interests.
