## [Peer Review File · Nature Communications]

REVIEWER COMMENTS

Reviewer #1 (Remarks to the Author):

In this manuscript by Dongyi Xu and colleagues, the authors identify ASF1 as an interacting partner of RIF1. They show that ASF1 acts in the same pathway as RIF1 in NHEJ but in a parallel pathway from Shieldin. They claim that ASF1 compacts chromatin adjacent to DNA DSBs to protect from BRCA1 mediated resection and its loss rescues the sensitivity of BRCA1 deficient cells.

Although the authors present an interesting concept which expands the ever-increasing role of RIF1 in cellular physiology, the some of the data presented show quite mild effects and the mechanisms proposed are not completely justified by the assays performed in the manuscript.

1. I am not sure of the choice of drugs to look into the effects of 53BP1, RIF1 and ASF1 in cellular viability. Why was etoposide used and not X rays which is the most commonly used DSB inducing DNA damage? TopII inhibitors cause transcription induced damage as well. How can the effects of transcription induced damage be ruled out?

2. I am not sure how the stripes have been quantified upon overexpression of GFP-ASF1. It is very difficult to see much from the representative images. The authors should consider staining with for endogenous ASF1 or probably use a knock-in as overexpression of these chromatin associated proteins can result on non-specific accumulation to sites of DNA damage. At the least the authors should test clones for overexpression and use the one closest to physiological levels for their assays.

3. I am not convinced by the conclusions drawn by the authors in Fig 2F and H where they say that ASF1 like 53BP1 and RIF1 accumulate distal to breaks. The data is at best mild where its very difficult to see any difference. ASF1 definitely does not show the same dynamics as 53BP1 or RIF1. I am very confused about this claim.

4. How does *asf1a*, *shld2* double knockout compare to *rif1* ko in terms of sensitivity?

5. In fig4. The authors quantify compaction of chromatin by measuring the area of ectopically expressed HP1 and H3K9me3. Both of them do not have site specificity and would be very hard to measure specifically at sites to DSBs without proper normalizations with the background levels. The authors should perform ChIP for these proteins and methylations at site specific DSBs to prove their point about condensation of chromatin at flanking regions.

6. Again in Fig.6 overexpression of Suv39h2 shows strange distribution of GFP in different cell types as shown in the representative pictures. Therefore, for these assays to have physiological relevance, either endogenous knock-in should be used or antibody stainings need to be performed.

Reviewer #2 (Remarks to the Author):

The MS "RIF1-ASF1-mediated high-order chromatin structure safeguards genome integrity" explores the role of conserved ASF1 histone chaperone in DNA double-strand break (DSB) repair by non-homologous end-joining (NHEJ). ASF1 has previously been implicated in facilitating ATM signaling, an upstream event required for 53BP1-RIF1 interaction and recruitment to DSBs, and therefore for efficient DNA end-protection and NHEJ repair. In a deviation from these previous results (ref. 68), here, an interesting interaction between ASF1 and NHEJ-promoting factor RIF1 is detected by IP/mass spec, which is subsequently verified by co-IP experiments and linked to a B domain-binding motif

within ASF1 depending on residues E36 and D37 that is critical for known ASF1 interactions with CAF-1 and HIRA. Thus, mutations E36A D37A (DEAA) in ASF1 abolished RIF1 binding by co-IP. Interestingly, these ASF1 mutations also abrogated GFP-ASF1 recruitment to laser-induced DNA damage in cells, and similar effects were observed after loss of 53BP1 or RIF1. Consistently, ASF1 was detected at the periphery of laser damage-induced BRCA1/53BP1-RIF1 foci in cells, and this localization was partially dependent on 53BP1 and RIF1. Mechanistically, ASF1 and RIF1 interactions are proposed to favour the recruitment of ASF1 to DNA double-strand breaks, where the actions of ASF1 may be required to mediate chromatin compaction (de-compaction microscopically observed in absence of ASF1) through monomethylated histone deposition, followed by SUV39h1/2 trimethylation and HP1-dependent heterochromatinization, thus limiting nuclease access (and perhaps mediating shieldin recruitment) to favour NHEJ repair. In contrast to the previous model of ASF1 promoting NHEJ through ATM signaling (ref. 68), the model presented here suggest a more direct (physical) link to RIF1 and chromatin compaction as a barrier to end-resection (referred to as the "53BP1-RIF1-ASF1 pathway").

The MS reports original results with the potential to advance the field by elucidating mechanistic insights into the multi-faceted involvement of 53BP1-RIF1 in DNA double-strand break pathway choice. The suggestion of a new way in which ASF1 may participate as an effector of chromatin compaction and DNA-end stability will be of interest to many in the field. To fully support the claims made in the MS, and make a more compelling case for their intriguing model, the authors should consider addressing the following points.

(1) Chicken DT40 cells are used to shown that ASF1, RIF1, and the interaction between ASF1 and RIF1 was required for resistance to DSB-inducing drugs, suggesting ASF1's requirement for NHEJ repair. However, the RIF1 mutant with B-domain mutations was able to promote significant rescue of rif1^{-/-} etoposide resistance while the RIF1-interaction-defective (also affecting other protein interactions) ASF1 mutant was unable to provide any rescue. Does ASF1 contribute significantly, perhaps mostly, to etoposide resistance (by proxy, NEHJ), along a pathway unrelated to any direct RIF1 physical interaction? Would this conform to the previous model of ref. 68 rather than the one presented here that is based on a direct ASF1-RIF1 protein-protein interaction?

(2) What is the evidence in human cells that ASF1 depletion suppresses PARPi sensitivity in BRCA1-depleted cells? Contrary to the statement (line 235) that depletion of ASF1 rescued the PARPi-sensitivity of BRCA1-deficient HEK293T cells (which would imply a compromised 53BP1-RIF1 axis and compromised end-protection), there is no evidence in Fig. 3i that this rescue occurs in human cells, while the evidence in DT40 cells (Fig. 3G, H) is more convincing. To support the claim, central to this MS, what other evidence (outside the DT40 system) can the authors supply?

(3) While Fig. 4 is titled "53BP1-RIF1-ASF1 condenses chromatin flanking DSBs", the assays presented address the proteins individually and, contrary to what the title implies, not as a joint pathway. It would be a major improvement to interrogate the intriguing physical interaction of ASF1 and RIF1 mediated by the RIF1 B domain (Fig. 1) directly in the DSB chromatin condensation setting. Only then can a compelling case for the RIF1-mediated ASF1 recruitment for chromatin compaction at DSBs be made (and if evidence cannot be provided, these claims should be revised). Unfortunately, currently, experiments to that effect have only been done in a non-DSB gain-of function scenario, which is much less compelling (Fig. 5).

(4) Discussion, line 352: "we identified ASF1 as a partner of RIF1 to protect broken ends". Should this statement be qualified as "in chicken DT40 cells"? What is the evidence for DSB end protection and promotion of NHEJ repair by ASF1 in mammalian cells?

(5) The discussion might benefit from a closer alignment with, and critical discussion of, the results presented. For example, a critical discussion of results from Figs. 4 and 5 is largely missing.

(6) Discussion, line 398: "The question of how the two functions of ASF1a in NHEJ are coordinated in different species and cells remain to be investigated in future." A more transparent discussion of the differences in observations in DT40 in the current MS (see e.g., point 2 above) and data from the literature in mammalian systems, and what the discrepancies might indicate, could be helpful, especially in light of the last Discussion section concerned with the relevance of the results presented for cancer.

Reviewer #3 (Remarks to the Author):

In this manuscript, Feng et al., identify the histone chaperone ASF1A as a partner of the DNA double-strand break (DSB) response protein RIF1 in human cells. Similar to ASF1 interaction with the histone chaperones CAF-1 and HIRA, ASF1 binding to RIF1 involves RIF1 B-domain. Using both human and chicken cellular models, the authors reveal that ASF1 works in the same pathway as RIF1, but in a parallel pathway to the shieldin complex, to inhibit DSB repair by homologous recombination (HR). RIF1 promotes the recruitment of ASF1A to DSB-flanking chromatin leading to the inhibition of BRCA1-dependent resection and facilitating DSB repair by non-homologous end joining (NHEJ). Mechanistically, this process involves ASF1 histone chaperone activity, and the authors provide evidence that ASF1 compacts chromatin around DSBs thus protecting DNA ends from resection.

This is an interesting, well-conducted, and clearly presented study, which provides novel mechanistic insights into the chromatin-based regulation of DSB repair pathways. Importantly, this work lends support to the hypothesis that the 53BP1-RIF1 axis strengthens the nucleosomal barrier to end resection. This work also raises important questions for future studies including a possible cooperation of ASF1 with RIF1 for the regulation of chromatin topology around DSBs. Several powerful tools were developed to study ASF1 function (knock-outs, Auxin-induced degradation, point mutations) across different species (human and chicken) and cellular models, thus strengthening the authors' findings. Nevertheless, the conclusions on ASF1-dependent heterochromatinization are far-fetched and not fully supported by the data presented. In addition, the authors should clarify discrepancies with the published literature regarding the role of ASF1A and H3K9me3 in repair by HR. The functional importance of RIF1 in competing out CAF-1 and HIRA for histone deposition has not been investigated, and important controls are missing, in particular regarding the cell cycle stage at which the experiments were performed. This work deserves being considered for publication in Nature Communications, should the authors appropriately address these points (as detailed below).

Major comments:

1. ASF1-dependent chromatin compaction and H3K9me3/HP1 recruitment to DSBs are observed at a very specific and artificial locus (LacO array) and only HP1 recruitment is confirmed at laser tracks. The validity of the findings genome-wide can thus be questioned and the claim about a "universal heterochromatinization" (line 279) is clearly an overstatement. Indeed, the recruitment of HP1 to DSBs is not necessarily a sign of heterochromatinization since it occurs in a CAF-1-dependent manner independently of H3K9me3 binding (PMID: 21464229). Thus, we suggest the authors to address whether ASF1 impacts HP1 recruitment via CAF-1 and to monitor H3K9me3 and chromatin compaction on laser tracks as well.

2. To test whether ASF1-dependent chromatin compaction close to DSBs (Fig. 4) involves histone deposition by ASF1 and binding to RIF1, the authors should express the ASF1 histone-binding mutant and RIF1 B-domain mutant. The data presented in Fig. 5 (non-DSB conditions) are not very relevant to the role of ASF1 post DSB. This figure could be removed or considerably reduced to better characterize the response to DSBs (by including ASF1 and RIF1 mutants in Fig. 4).

3. The recruitment of ASF1 to broken chromatin is assessed at laser tracks and at FokI cut sites, but is

only shown with the GFP-tagged protein. The recruitment of endogenous ASF1 should be examined as well (by immunofluorescence using ASF1A/B antibodies for example). In addition, showing that ASF1A recruitment to the periphery of laser damage spots (Fig. 2) is abolished upon expression of the RIF1 mutant form would support the conclusion that RIF1 drives ASF1 accrual on chromatin distal to DSBs.

4. ASF1A loss-of-function partially restores the sensitivity to Olaparib in BRCA1-deficient cells (Fig. 3). The histone binding mutant of ASF1A and the RIF1 B-domain mutant should be included in these analyses. This would strengthen the data presented and support the conclusion that the chaperone activity of ASF1 and its binding to RIF1 are critical for the suppression of HR.

5. Since HR is mainly observed in S and G2 phases of the cell cycle, the authors should control the cell cycle distribution when monitoring HR activity and HR factor recruitment.

6. The authors should clarify the discrepancies with the published literature regarding the role of ASF1A and H3K9me3 in HR. Indeed, while in this study ASF1A histone chaperone activity inhibits HR at the resection level, it was shown to stimulate HR at the RAD51 loading step in a previous study (PMID: 29478807). In addition, H3K9me3 and HP1 channel DSB repair towards HR (PMID: 26206670) and a local increase in H3K9me3 is important to stimulate HR (PMID: 32494005) while here the authors find the opposite. The authors argue that these differences could be explained by a temporal separation of events (p.14 and supplementary fig. 9). This is indeed an interesting hypothesis but the authors should provide data such as kinetic analyses to support these assumptions. For example, is the impact of ASF1A on DSB repair by HR dependent on the timing post DSB induction? Is RIF1-dependent recruitment of ASF1 to laser tracks detectable only at specific time points post damage? In this respect, the timing post laser damage should be indicated for all laser experiments.

7. CAF-1, HIRA, and RIF1 all bind to ASF1 through their B-domain. This suggests that RIF1 could compete with CAF-1 and HIRA for binding ASF1, thus impeding CAF-1- and HIRA-mediated histone deposition pathways post damage. This should be tested in the present work.

Minor comments:

- Fig. 1: Is 53BP1 part of the RIF1-ASF1 complex (Fig. 1A-C)?

RIF1 binds to chromatin post damage. May this explain why more H3 and ASF1 are retrieved in the RIF1 pull-down (Fig. 1B)?

The IP shown in Fig. 1C should be performed +/- Bleomycin as it is the case in Fig. 1B.

The authors should clarify why ASF1B is not retrieved in RIF1 pull-down (Fig. 1B) but is recruited to laser tracks in a RIF1-dependent manner (Fig. 1H).

- Supp. Fig. 2: it is surprising not to detect H3 in HIRA and CAF-1 pull-downs.

- The authors should provide a control WB for RIF1 KO (Fig.1), and should control the effect of 53BP1 and RIF1 KO on ASF1 total levels since an upregulation of ASF1 could mask ASF1 local enrichment on laser tracks (Fig. 1F-G). Similarly, the KO of *rif1*, *brca1*, and *shld2* genes in DT40 cells should be verified (Supp. Fig. 1). In Fig. 6, the authors should control the effect of 53BP1-, RIF1- and ASF1-KO on the total levels of Suv39, and should assess whether the depletion of Suv39 impacts the total levels of BRCA1 and RPA. Indeed, knocking-down a transcriptional repressor may affect the expression of many genes.

- Fig. 6E: siSUV39H1/2 leads to a global decrease in H3K9me3, as expected. Therefore, it is difficult to conclude about a local decrease of this mark at DSBs.

- Suppl. Fig. 1: the effects of shieldin and *rif1* loss are compared in chicken DT40 cells. Could the authors state whether paralogs of *shld2* exists in chicken, which might explain why the *shieldin2* KO is

not as effective as the RIF1 KO in rescuing the BRCA1 deficiency in response to PARPi.

- Line 179: the authors should rephrase the sentence stating that ASF1A recruitment to the core region was not significantly affected by the disruption of 53BP1 or RIF1 because Fig. 2I shows an increase of at least 2-fold of ASF1A recruitment to DSB-proximal regions in both mutants compared to control cells.

- Laser micro-irradiation has long been recognized as a useful tool to assess the spatial distribution of repair factor recruitment to DSBs (PMID: 16618811). Thus, the HIRDC method is not strikingly novel. Moreover, the authors should specify in the method section which laser wavelength is used for laser micro-irradiation experiments. Surprisingly, it seems that the standard and high-energy laser experiments both use 65% laser output, while a higher laser output would be expected for the high-energy conditions.

- Fig. 2J, the minimal number of cells scored per condition should be indicated. Regarding statistics, one-way ANOVA should be used instead of t-tests when comparing more than two samples. This applies to all statistical tests in this study. Moreover, it would be helpful to include statistics in Fig. 1I, and in Fig. 6L (brca1-/- vs brca1-/-suv39h-/- survival to Olaparib).

- The authors should explain why they analyzed two HR markers, RPA32 and RAD51, in different cell lines, chicken DT40 and human U2OS (Fig. 3K, Suppl. Fig. 5A).

- The authors should justify the focus on HP1gamma (as opposed to alpha and beta).

- Fig. 4C: the authors could specify that they measure mean intensities (bottom panels), which are insensitive to changes in area (top panel).

- Final model in Fig. 7: In order to simplify the message, we recommend that the authors only show the panel A.

Discussion points:

- Both ASF1 and RIF1 are conserved in budding yeast, thus, whether the role of ASF1 in regulating chromatin compaction and resection at DSBs is evolutionary conserved could be discussed.

- Line 365: What would support the assumption that RIF1 may have a histone chaperone activity? Does RIF1 harbor an acidic domain that may contribute to histone binding?

- Lines 447-448: Note that the combination of HDAC inhibitors and PARP inhibitors has already been tested and shown efficacy for the treatment of breast cancer (PMID: 25026298) and glioblastoma for example (PMID: 26794465).

Typographical errors:

- Line 76, the title of the first result section needs rephrasing.

- Line 325, "meditated" instead of "mediated".

- Line 370, a verb is missing at the beginning of the sentence.

- Lines 386 and 433, replace "ensure" by "ensue".

- Line 445, replace "HADC" by "HDAC".

- Fig. 3E, the red curve should be labeled asf1a-/-+.

REVIEWER COMMENTS

Reviewer #1 (Remarks to the Author):

In this manuscript by Dongyi Xu and colleagues, the authors identify ASF1 as an interacting partner of RIF1. They show that ASF1 acts in the same pathway as RIF1 in NHEJ but in a parallel pathway from Shieldin. They claim that ASF1 compacts chromatin adjacent to DNA DSBs to protect from BRCA1 mediated resection and its loss rescues the sensitivity of BRCA1 deficient cells.

Although the authors present an interesting concept which expands the ever-increasing role of RIF1 in cellular physiology, the some of the data presented show quite mild effects and the mechanisms proposed are not completely justified by the assays performed in the manuscript.

Re: We thank this reviewer for his/her comments and suggestions.

1. I am not sure of the choice of drugs to look into the effects of 53BP1, RIF1 and ASF1 in cellular viability. Why was etoposide used and not X rays which is the most commonly used DSB inducing DNA damage? TopII inhibitors cause transcription induced damage as well. How can the effects of transcription induced damage be ruled out?

Re: We agree that X rays are commonly used to induce DSBs in many cells in many experiments. X rays usually induce “dirty” broken ends, which tends to be repaired by HR (PMID: 32648897; PMID: 21317870), although both NHEJ and HR can repair such DSBs. DT40 cells have much stronger HR repair activity than most cell lines, as evidenced by reports showing that DT40 cells have very high gene targeting efficiency in classic gene-knockout strategies. Therefore, NHEJ-deficient DT40 cells, such as *53BP1*^{-/-} and *RIF1*^{-/-}DT40 cells, usually are not (or not very) sensitive to X rays (PMID: 16644291; PMID: 16866876; PMID: 23333306), and asynchronous *Ku70*^{-/-} DT40 cells are more resistant to X rays than wild-type cells at high doses (PMID: 9736627; PMID: 16866876). Therefore, X rays are not suitable for our experiments in DT40 cells.

In contrast, etoposide-induced DSB ends are usually “clean” and tend to be repaired by NHEJ (PMID: 21317870; PMID: 25670504). It has been reported by many groups that NHEJ-deficient cells, including NHEJ-deficient DT40 cells, are hypersensitive to etoposide and other TopII inhibitors (PMID: 12842886; PMID: 16644291; PMID: 25670504). Therefore, we selected etoposide as the major drug to induce DSBs.

Cells with defects in any NHEJ core factors (such as Ku70/80, XRCC4, Lig4, XLF and PAXX) or any NHEJ regulators in the 53BP1 pathway (such as 53BP1, RIF1 and Shieldin) are hypersensitive to etoposide (PMID: 12842886; PMID: 16644291; PMID: 25670504; PMID: 25670504; PMID: 30254264). Those genes also are epistatic in resistance to etoposide (PMID: 25670504; PMID: 30254264). These findings demonstrate that the NHEJ pathway is required to repair etoposide-induced damage. It is well established that the major function of NHEJ is DSB repair, and, therefore, etoposide-induced DSBs, but not others, are the main cause for the death of NHEJ-deficient cells. We cannot exclude the possibility that transcription-induced damages by etoposide indirectly

cause DSBs. In addition, we executed foreign DNA random integration assays, for which NHEJ is essential (PMID: 23333306; PMID: 30254264), to confirm the function of ASF1a and RIF1 in NHEJ (Figure 3b).

2. I am not sure how the stripes have been quantified upon overexpression of GFP-ASF1. It is very difficult to see much from the representative images. The authors should consider staining with for endogenous ASF1 or probably use a knock-in as overexpression of these chromatin associated proteins can result on non-specific accumulation to sites of DNA damage. At the least the authors should test clones for overexpression and use the one closest to physiological levels for their assays. Re: We thank this reviewer for his/her suggestions. We executed these experiments by staining with endogenous ASF1 as suggested. Similar to GFP-ASF1a, endogenous ASF1a is recruited to DSB sites by 53BP1 and RIF1 (Supplementary Fig. 3b-d). Because the anti-ASF1b antibodies could not recognize endogenous ASF1b well in immune staining (see below), we could not determine whether endogenous ASF1b was recruited to laser-induced DNA damage sites as well.

Figure legend: a, Immuno-staining of ASF1b after laser-induced DNA damage. b, Immuno-staining of ASF1b in wild-type and *ASF1b*-null HEK293T cells. The ASF1b staining signal was not decreased in *ASF1b*-null cells, suggesting that the antibodies could not recognize endogenous ASF1b well.

3. I am not convinced by the conclusions drawn by the authors in Fig 2F and H where they say that ASF1 like 53BP1 and RIF1 accumulate distal to breaks. The data is at best mild where its very difficult to see any difference. ASF1 definitely does not show the same dynamics as 53BP1 or RIF1. I am very confused about this claim.

Re: We apologize that we did not explain the data well. We agree with the reviewer that ASF1 does not show the same dynamics as 53BP1 or RIF1. 53BP1 and RIF1 only

accumulate to distal to breaks, while ASF1 accumulates to both distal and proximal to breaks.

We define the region of ssDNA and proximal dsDNA to breaks (the center region induced by laser) using mCherry-MRE11. The region outside of mCherry-MRE11 is defined as distal dsDNA to breaks. Under the conditions in Fig2b-h, 53BP1 and RIF1 only accumulate outside of the region of mCherry-MRE11, indicating that they only accumulate distal to breaks. Similar to 53BP1 and RIF1, ASF1 also accumulated outside of the region of mCherry-MRE11 (Fig f, g), and this accumulation was impaired in 53BP1 or RIF1-deficient cells (Fig h-j), suggesting that ASF1 is recruited to regions distal to breaks by 53BP1 and RIF1.

Different to 53BP1 and RIF1, ASF1 can also accumulate in the region with mCherry-MRE11. This accumulation of ASF1 may be mediated by other proteins, such as HR factors, because ASF1 was reported to play a role in HR (PMID: 29478807).

4. How does *asf1a*, *shld2* double knockout compare to *rif1* ko in terms of sensitivity?

Re: We thank the reviewer for this suggestion. We generated *asf1a/shld2/rif1* triple knockout cells and compared the drug sensitivity of those cell lines. *Rif1*^{-/-}, *asf1*^{-/+shld2}^{-/-} and *rif1*^{-/-}*asf1*^{-/+shld2}^{-/-} cells showed similar sensitivity to ICRF193 (Fig. 3h), which induces DSBs that are specifically repaired by NHEJ (PMID: 20088963). This result is consistent with our model, in which ASF1a and Shieldin act at two parallel pathways downstream of RIF1 to promote NHEJ.

In contrast to ICRF193 sensitivity, *rif1*^{-/-}*asf1*^{-/+shld2}^{-/-} cells are slightly more sensitive than *rif1*^{-/-} or *asf1*^{-/+shld2}^{-/-} cells to etoposide, which induces DSBs that can be repaired by both NHEJ and HR (PMID: 20088963) (Fig. 3h), suggesting that ASF1a, Shieldin and RIF1 may have functions other than NHEJ.

5. In fig4. The authors quantify compaction of chromatin by measuring the area of ectopically expressed HP1 and H3K9me3. Both of them do not have site specificity and would be very hard to measure specifically at sites to DSBs without proper normalizations with the background levels. The authors should perform CHIP for these proteins and methylations at site specific DSBs to prove their point about condensation of chromatin at flanking regions.

Re: We apologize that we did not describe the data and method well. The area of the LacO array, which is bound by mCherry-LacR, was measured by measuring the area of mCherry in living cells using Imaris software. The HP1 and H3K9me3 signals were measured using ImageJ as described below. In this method, the LacO-array region is defined very accurately after magnification, and the dot of the LacO-array region is very small, so that other signals can be excluded efficiently. In addition, the final value is strictly normalized with the nucleus and background signal (see below). In our laboratory, the results using this method are more reproducible than those obtained with CHIP. We also performed CHIP of H3K9me3 as suggested (Fig. 5d), and the results were consistent with those obtained using the U2OS-265 system.

Workflow for the quantification of HP1 or H3K9me3 at the LacO-array:

- ① Open the images using ImageJ software; magnify the zone containing a LacO-array in the mCherry (red) channel. Define the region of the LacO-array (encircle the dot of the mCherry signal as closely as possible using “freehand selections”).
- ② In the same window, switch to the green channel (H3K9me3 or GFP-CBX3) and the yellow circle, which marks the defined region of the LacO-array, will be retained. The mean intensity of H3K9me3 or GFP-CBX3 in the LacO-array region (I_{array}) can be measured.
- ③ In the green channel, resize the image to fit, define the region of the nucleus and measure the mean intensity of H3K9me3 or GFP-CBX3 in the nucleus (I_{nu}).
- ④ In the same window, select an arbitrary region without cells as the background region (I_b). Measure the mean intensity of I_b .
- ⑤ Calculate the related intensity of H3K9me3 or GFP-CBX3 of the array:

$$I = (I_{array} - I_{nu}) / (I_{nu} - I_b)$$

6. Again in Fig.6 overexpression of Suv39h2 shows strange distribution of GFP in different cell types as shown in the representative pictures. Therefore, for these assays to have physiological relevance, either endogenous knock-in should be used or antibody stainings need to be performed.

Re: We thank the reviewer for this suggestion. We performed endogenous Suv39h1 staining as suggested and obtained similar results (Fig. 6a, b). The anti-Suv39h2 antibodies are not suitable for immune-staining (data not shown).

Reviewer #2 (Remarks to the Author):

The MS “RIF1-ASF1-mediated high-order chromatin structure safeguards genome integrity” explores the role of conserved ASF1 histone chaperone in DNA double-strand break (DSB) repair by non-homologous end-joining (NHEJ). ASF1 has previously been implicated in facilitating ATM signaling, an upstream event required for 53BP1-RIF1 interaction and recruitment to DSBs, and therefore for efficient DNA end-protection and NHEJ repair. In a deviation from these previous results (ref. 68), here, an interesting interaction between ASF1 and NHEJ-promoting factor RIF1 is detected by IP/mass spec, which is subsequently verified by co-IP experiments and linked to a B domain-binding motif within ASF1 depending on residues E36 and D37 that is critical for known ASF1 interactions with CAF-1 and HIRA. Thus, mutations E36A D37A (DEAA) in ASF1 abolished RIF1 binding by co-IP. Interestingly, these ASF1 mutations also abrogated GFP-ASF1 recruitment to laser-induced DNA damage in cells, and similar effects were observed after loss of 53BP1 or RIF1. Consistently, ASF1 was detected at the periphery of laser damage-induced BRCA1/53BP1-RIF1 foci in cells, and this localization was partially dependent on 53BP1 and RIF1. Mechanistically, ASF1 and RIF1 interactions are proposed to favour the recruitment of ASF1 to DNA double-strand breaks, where the actions of ASF1 may be required to mediate chromatin compaction (de-compaction microscopically observed in absence of ASF1) through monomethylated histone deposition, followed by SUV39h1/2 trimethylation and HP1-dependent heterochromatinization, thus limiting nuclease access (and perhaps mediating shieldin recruitment) to favour NHEJ repair. In contrast to the previous model of ASF1 promoting NHEJ through ATM signaling (ref. 68), the model presented here suggest a more direct (physical) link to RIF1 and chromatin compaction as a barrier to end-resection (referred to as the “53BP1-RIF1-ASF1 pathway”).

The MS reports original results with the potential to advance the field by elucidating mechanistic insights into the multi-faceted involvement of 53BP1-RIF1 in DNA double-strand break pathway choice. The suggestion of a new way in which ASF1 may participate as an effector of chromatin compaction and DNA-end stability will be of interest to many in the field. To fully support the claims made in the MS, and make a more compelling case for their intriguing model, the authors should consider addressing the following points.

Re: We thank this reviewer for his/her comments and suggestions.

(1) Chicken DT40 cells are used to shown that ASF1, RIF1, and the interaction between ASF1 and RIF1 was required for resistance to DSB-inducing drugs, suggesting ASF1’s requirement for NHEJ repair. However, the RIF1 mutant with B-domain mutations was able to promote significant rescue

of *rif1*^{-/-} etoposide resistance while the RIF1-interaction-defective (also affecting other protein interactions) ASF1 mutant was unable to provide any rescue. Does ASF1 contribute significantly, perhaps mostly, to etoposide resistance (by proxy, NEHJ), along a pathway unrelated to any direct RIF1 physical interaction? Would this conform to the previous model of ref. 68 rather than the one presented here that is based on a direct ASF1-RIF1 protein-protein interaction?

Re: We thank the reviewer for this comment. In addition to etoposide-sensitivity assays, we have now performed two additional complementary experiments to examine NHEJ using the B-domain-mutated RIF1 (RIF1_AAD). First, we carried out the foreign DNA random integration assay, which is exclusively dependent on NHEJ in DT40 cells, and found that RIF1_AAD fully lost its function to promote random integration (Fig. 3d). Second, RIF1_AAD fully lost its function to confer PARPi-sensitivity on BRCA1-deficient cells (Fig. 4g, h). This function of RIF1 is dependent on the 53BP1 pathway to antagonize BRCA1-mediated resection and HR. Therefore, the ASF1-RIF1 interaction is important for their roles in antagonizing BRCA1 and promoting NHEJ.

Consistent with this conclusion, RIF1_AAD could not rescue the etoposide sensitivity of *rif1*^{-/-} cells as effectively as wild-type RIF1, although it was partially rescued. So far, the mechanism underlying the partially rescued function of RIF1_AAD in the etoposide-sensitivity assay is unclear. One possibility is that the RIF1_AAD mutant can still partially resist etoposide through its interaction with Shieldin, although it does not influence random integration and antagonizing BRCA1 for unknown reasons. We added this discussion in the new manuscript.

(2) What is the evidence in human cells that ASF1 depletion suppresses PARPi sensitivity in BRCA1-depleted cells? Contrary to the statement (line 235) that depletion of ASF1 rescued the PARPi-sensitivity of BRCA1-deficient HEK293T cells (which would imply a compromised 53BP1-RIF1 axis and compromised end-protection), there is no evidence in Fig. 3i that this rescue occurs in human cells, while the evidence in DT40 cells (Fig. 3G, H) is more convincing. To support the claim, central to this MS, what other evidence (outside the DT40 system) can the authors supply?

Re: We agree with this reviewer that the difference is not very strong between these cells in the previous Fig. 3i because the PARPi sensitivity of the BRCA1-depleted HEK293T cells was not very strong. We changed to another mammalian cell line HCT116 cells, in which disrupting the BRCA1 gene generates strong PARPi sensitivity. The quality of the new figure is significantly improved (Fig. 4c).

(3) While Fig. 4 is titled “53BP1-RIF1-ASF1 condenses chromatin flanking DSBs”, the assays presented address the proteins individually and, contrary to what the title implies, not as a joint pathway. It would be a major improvement to interrogate the intriguing physical interaction of ASF1 and RIF1 mediated by the RIF1 B domain (Fig. 1) directly in the DSB chromatin condensation setting. Only then can a compelling case for the RIF1-mediated ASF1 recruitment for chromatin compaction at DSBs be made (and if evidence cannot be provided, these claims should be revised). Unfortunately, currently, experiments to that effect have only been done in a non-DSB gain-of-function scenario, which is much less compelling (Fig. 5).

Re: We thank the reviewer for this suggestion. We performed complementary

experiments in those assays. Wild-type forms, but not the interacting-domain-mutated ASF1 or RIF1, condensed chromatin flanking DSBs. These results were included in the new manuscript (Fig. 5e-h).

(4) Discussion, line 352: “we identified ASF1 as a partner of RIF1 to protect broken ends”. Should this statement be qualified as “in chicken DT40 cells”? What is the evidence for DSB end protection and promotion of NHEJ repair by ASF1 in mammalian cells?

Re: We added “in chicken DT40 cells” in this statement as suggested.

(5) The discussion might benefit from a closer alignment with, and critical discussion of, the results presented. For example, a critical discussion of results from Figs. 4 and 5 is largely missing.

Re: We thank the reviewer for this suggestion. We rearranged the discussion and added discussion contents after the results of Fig. 5 (Fig. 4 in the previous version). Fig. 5 in the previous version was removed as suggested by reviewer #3.

(6) Discussion, line 398: “The question of how the two functions of ASF1a in NHEJ are coordinated in different species and cells remain to be investigated in future.” A more transparent discussion of the differences in observations in DT40 in the current MS (see e.g., point 2 above) and data from the literature in mammalian systems, and what the discrepancies might indicate, could be helpful, especially in light of the last Discussion section concerned with the relevance of the results presented for cancer.

Re: As suggested, we added a more transparent and detailed discussion about the discrepancies between our data and the literature in mammalian systems in the new version of the manuscript.

Reviewer #3 (Remarks to the Author):

In this manuscript, Feng et al., identify the histone chaperone ASF1A as a partner of the DNA double-strand break (DSB) response protein RIF1 in human cells. Similar to ASF1 interaction with the histone chaperones CAF-1 and HIRA, ASF1 binding to RIF1 involves RIF1 B-domain. Using both human and chicken cellular models, the authors reveal that ASF1 works in the same pathway as RIF1, but in a parallel pathway to the shieldin complex, to inhibit DSB repair by homologous recombination (HR). RIF1 promotes the recruitment of ASF1A to DSB-flanking chromatin leading to the inhibition of BRCA1-dependent resection and facilitating DSB repair by non-homologous end joining (NHEJ). Mechanistically, this process involves ASF1 histone chaperone activity, and the authors provide evidence that ASF1 compacts chromatin around DSBs thus protecting DNA ends from resection.

This is an interesting, well-conducted, and clearly presented study, which provides novel mechanistic insights into the chromatin-based regulation of DSB repair pathways. Importantly, this work lends support to the hypothesis that the 53BP1-RIF1 axis strengthens the nucleosomal barrier to end resection. This work also raises important questions for future studies including a possible cooperation of ASF1 with RIF1 for the regulation of chromatin topology around DSBs. Several

powerful tools were developed to study ASF1 function (knock-outs, Auxin-induced degradation, point mutations) across different species (human and chicken) and cellular models, thus strengthening the authors' findings.

Nevertheless, the conclusions on ASF1-dependent heterochromatinization are far-fetched and not fully supported by the data presented. In addition, the authors should clarify discrepancies with the published literature regarding the role of ASF1A and H3K9me3 in repair by HR. The functional importance of RIF1 in competing out CAF-1 and HIRA for histone deposition has not been investigated, and important controls are missing, in particular regarding the cell cycle stage at which the experiments were performed. This work deserves being considered for publication in Nature Communications, should the authors appropriately address these points (as detailed below).

Re: We thank this reviewer for his/her recommendation and suggestions. We modified the manuscript as suggested (please see below).

Major comments:

1. ASF1-dependent chromatin compaction and H3K9me3/HP1 recruitment to DSBs are observed at a very specific and artificial locus (LacO array) and only HP1 recruitment is confirmed at laser tracks. The validity of the findings genome-wide can thus be questioned and the claim about a "universal heterochromatinization" (line 279) is clearly an overstatement. Indeed, the recruitment of HP1 to DSBs is not necessarily a sign of heterochromatinization since it occurs in a CAF-1-dependent manner independently of H3K9me3 binding (PMID: 21464229). Thus, we suggest the authors to address whether ASF1 impacts HP1 recruitment via CAF-1 and to monitor H3K9me3 and chromatin compaction on laser tracks as well.

Re: We thank this reviewer for his/her comments. We changed the claim to "These results suggest that 53BP1-RIF1-ASF1-mediated heterochromatinization may occur at common DSB sites".

As mentioned by this reviewer, HP1 was reported to be recruited to DSBs by CAF-1 p150, but independently of H3K9me3 binding, Suv39H1/2 and CAF-1 p60 (PMID: 21464229). As suggested by the reviewer, we examined whether ASF1 impacts HP1 recruitment via CAF-1. In our system (HEK293T cells and Micro-point with a 365 nm laser), we did not observe significant differences in HP1 γ recruitment after depletion of CAF-1 p150 or p60 in the wild-type or *ASF1a^dASF1b^H* HEK293T cells (Supplementary Fig. 11a-f), suggesting that CAF-1 is not important for ASF1-dependent recruitment of HP1 in our system. In contrast, we generated Suv39H1/2 double knockout HEK293T cells, and we found that HP1 γ recruitment to DSBs was impaired in the Suv39H1/2 double knockout cells (~30% decrease compared with that of the wild-type cells; Supplementary Fig. 12a-c). Therefore, at least a subset of HP1 is recruited to DSBs by Suv39H1/2 and/or Suv39H1/2-mediated H3K9me3. Our results are consistent with a previous study reporting that loading of Suv39h1, Kap-1, and HP1 at DSBs is interdependent and leads to cycles and spreading of Kap-1-HP1-Suv39h1 loading and H3K9 methylation along the chromatin (PMID: 24927542). The discrepancies between our results and the former published study (PMID: 21464229) may be due to the use

of different DSB induction systems (405 nm laser by a confocal microscope with Hoechst vs. 365nm laser by a Micro-point) or cell lines.

Although H3K9me3 was found to be recruited to DSB sites by CHIP assay (PMID: 24927542), previous studies did not detect obvious accumulation of H3K9me3 to laser-induced damage tracks (PMID: 21464229; PMID: 30312172). Most recently, accumulation of H3K9me3 at DSB sites was clearly observed by both CHIP and a microlaser-based method (PMID: 32494005). In our system, we only detected very weak accumulation of H3K9me3 signal at laser tracks in a few U2OS and MEF cells (less than 20% of cells; see figures below). The signal was too weak for quantification. The poor-quality of the signal at the laser tracks was likely due to the high background of H3K9me3 in the nucleus. This is the reason why we used the U2OS-265 system to examine H3K9me3 accumulation. In the U2OS-265 cell system, the LacO array with FokI-induced DSBs is present as a small dot, and it is marked by mCherry-LacR. These small dots can be distinguished easily from the background (see the workflow in the response to point 5 of reviewer #1). Therefore, measuring the H3K9me3 signal of DSB-contained LacO-array dots is a more sensitive method in comparison with the laser-track-based procedure.

Figure legends: Representative images of H3K9me3 staining after laser-induced DNA damage in U2OS (a) and MEF (b) cells. A few cells (<20%) show weak accumulated H3K9me3 signal at the laser tracks.

2. To test whether ASF1-dependent chromatin compaction close to DSBs (Fig. 4) involves histone deposition by ASF1 and binding to RIF1, the authors should express the ASF1 histone-binding mutant and RIF1 B-domain mutant. The data presented in Fig. 5 (non-DSB conditions) are not very relevant to the role of ASF1 post DSB. This figure could be removed or considerably reduced to better characterize the response to DSBs (by including ASF1 and RIF1 mutants in Fig. 4).

Re: We thank the reviewer for this suggestion. We performed complementation experiments as suggested and found that both interaction domains of ASF1a and RIF1 are required for their functions in promoting heterochromatinization near DSBs (Fig.

5e-h). In addition, the histone chaperone (histone binding) activity of ASF1a is important for this function (Fig. 5e,f). These data were included in the new manuscript. Moreover, we removed the data obtained under non-DSB conditions in the previous Fig. 5 as suggested.

3. The recruitment of ASF1 to broken chromatin is assessed at laser tracks and at FokI cut sites, but is only shown with the GFP-tagged protein. The recruitment of endogenous ASF1 should be examined as well (by immunofluorescence using ASF1A/B antibodies for example). In addition, showing that ASF1A recruitment to the periphery of laser damage spots (Fig. 2) is abolished upon expression of the RIF1 mutant form would support the conclusion that RIF1 drives ASF1 accrual on chromatin distal to DSBs.

Re: We performed these experiments by staining with endogenous ASF1 as suggested. Endogenous ASF1a is recruited to DSB sites by 53BP1 and RIF1, similar to GFP-tagged ASF1a (Supplementary Fig.3 b-d). In addition, the anti-ASF1b antibodies could not recognize ASF1b well in the immunofluorescence experiments (see the response to point 2 by reviewer #1).

We generated a RIF1_B-domain-mutated cell line and found that ASF1a accumulation on chromatin distal to DSBs was reduced in these cells (Fig. 2k and Supplementary Fig. 5a, b), suggesting that RIF1 drives ASF1 accumulation on chromatin distal to DSBs.

4. ASF1A loss-of-function partially restores the sensitivity to Olaparib in BRCA1-deficient cells (Fig. 3). The histone binding mutant of ASF1A and the RIF1 B-domain mutant should be included in these analyses. This would strengthen the data presented and support the conclusion that the chaperone activity of ASF1 and its binding to RIF1 are critical for the suppression of HR.

Re: We thank the reviewer for this suggestion. We performed complementation experiments as suggested. The RIF1 B-domain mutant (RIF1-AAD) lost its ability to confer olaparib-sensitivity on BRCA1-deficient cells (Fig. 4g, h), suggesting that the interaction of RIF1 with ASF1 is critical for its function in suppression of HR. We included these results in the new manuscript.

Although we made a great effort, we did not generate the *brca1^{-/-}asf1a^{-/-}/AID* DT40 cells complemented with ASF1a_V94R. *brca1^{-/-}asf1a^{-/-}/AID* DT40 cells grow worse than *brca1^{-/-}* cells or *asf1a^{-/-}/AID* cells. Moreover, ASF1a_V94R has dominant negative effect and the *asf1a^{-/-}/AID* cells complemented with ASF1a_V94R grow worse than *asf1a^{-/-}/AID* cells. Therefore, expression of ASF1a_V94R in *brca1^{-/-}asf1a^{-/-}/AID* DT40 cells may be lethal or very severe.

5. Since HR is mainly observed in S and G2 phases of the cell cycle, the authors should control the cell cycle distribution when monitoring HR activity and HR factor recruitment.

Re: We examined the cell cycle distribution of BRCA1 or/and ASF1 deficient DT40 and U2OS cells. Acute depletion of ASF1 and/or BRCA1 did not change the cell cycle in U2OS cells (Supplementary Fig. 7d). Depletion of ASF1a in BRCA1-deficient DT40 cells caused an increased population in the G1 phase (supplementary Fig. 7e). This effect

was unlikely to lead to an increase in RAD51 (HR) in *ASF1 α ^{-/-}/AID**BRCA1^{-/-}* cells compared with *BRCA1^{-/-}* cells because the G1 phase suppresses HR.

Moreover, as shown in Fig. 5F, G of the previous manuscript, RPA32 accumulation in U2OS-265 cells system was examined in the G1 phase. In the new manuscript, we also examined BRCA1 accumulation using G1 phase cells (Fig. 6f, h and Supplementary Fig. 9a, b).

6. The authors should clarify the discrepancies with the published literature regarding the role of ASF1A and H3K9me3 in HR. Indeed, while in this study ASF1A histone chaperone activity inhibits HR at the resection level, it was shown to stimulate HR at the RAD51 loading step in a previous study (PMID: 29478807). In addition, H3K9me3 and HP1 channel DSB repair towards HR (PMID: 26206670) and a local increase in H3K9me3 is important to stimulate HR (PMID: 32494005) while here the authors find the opposite. The authors argue that these differences could be explained by a temporal separation of events (p.14 and supplementary fig. 9). This is indeed an interesting hypothesis but the authors should provide data such as kinetic analyses to support these assumptions. For example, is the impact of ASF1A on DSB repair by HR dependent on the timing post DSB induction? Is RIF1-dependent recruitment of ASF1 to laser tracks detectable only at specific time points post damage? In this respect, the timing post laser damage should be indicated for all laser experiments.

Re: As we mentioned in the Discussion, these discrepancies regarding the functions of heterochromatin marks on HR have been reported in previous studies for several years. For example, two studies found that HP1 and KAP1 trigger inhibition of DNA end resection and HR (PMID:25818296; PMID:25905708), while another study showed that they promote HR (PMID: 26206670). Moreover, Tip60 is recruited by heterochromatin and H3K9me3 (PMID: 19783983). Meanwhile, Tip60 decondenses chromatin through H4 acetylation and promotes HR (PMID: 23377543). Several studies have shown that DSBs that occur in the more open, active chromatin environment of euchromatin are readily repaired by HR, while those occurring in closed, repressive heterochromatin are generally repaired by NHEJ (PMID:25818296; PMID: 25366693; PMID: 24658350). Brendan Price proposed a model to explain those discrepancies by a temporal separation of events (PMID: 26625977). In our manuscript, we modified this model by adding our data and other new findings (including the finding of a transient expansion less than 1.5 min after DSB). The Price's model was cited, and we emphasize that our hypothesis is modified from his model in the new manuscript.

We thank this reviewer for his/her suggestions to examine this model. Unfortunately, the previous study did not figure out which protein recruits ASF1 to DSB sites to promote HR (PMID: 29478807). Therefore, we could not examine when ASF1 is recruited by its upstream HR factors.

We examined the recruitment of ASF1 to laser tracks with a prolonged time period (up to 2 hr) as suggested (Supplementary Fig. 3c, d). We found that ASF1 recruitment was strongly reduced in the early period after DNA damage when RIF1 or 53BP1 is

disrupted. This result is consistent with our hypothesis that RIF1-dependent recruitment of ASF1 happens at the early stage of DSB for NHEJ repair. However, we could not explain why the recruitment of ASF1 in 53BP1- or RIF1-deficient cells did not recover to the level of the wild-type cells even after 2 hr. One possibility is that the recruitment of ASF1 by 53BP1-RIF1 in the early stage is also helpful at the late stage for HR, as we hypothesized in the Discussion. More studies using new strategies and techniques are necessary to test this hypothesis in the future.

In addition, we indicated the timing after laser damage for all laser experiments in the new manuscript.

7. CAF-1, HIRA, and RIF1 all bind to ASF1 through their B-domain. This suggests that RIF1 could compete with CAF-1 and HIRA for binding ASF1, thus impeding CAF-1- and HIRA-mediated histone deposition pathways post damage. This should be tested in the present work.

Re: We examined their interactions after DNA damage induced by bleomycin. The interaction of ASF1 with RIF1 was increased after bleomycin treatment (Fig. 1b, c), while the interactions of ASF1 with CAF-1 and HIRA were not affected (Supplementary Fig. 2b, c). Therefore, it is unlikely that RIF1 can compete with CAF-1 and HIRA for binding with ASF1 after DNA damage, impeding the CAF-1- and HIRA-mediated histone deposition pathways. It is possible that ASF1 is adequate for binding by all those proteins, or RIF1 only binds a small part of ASF1 (for example, only those H3K9me1-bound ASF1).

Minor comments:

- Fig. 1: Is 53BP1 part of the RIF1-ASF1 complex (Fig. 1A-C)?

RIF1 binds to chromatin post damage. May this explain why more H3 and ASF1 are retrieved in the RIF1 pull-down (Fig. 1B)?

The IP shown in Fig. 1C should be performed +/- Bleomycin as it is the case in Fig. 1B.

The authors should clarify why ASF1B is not retrieved in RIF1 pull-down (Fig. 1B) but is recruited to laser tracks in a RIF1-dependent manner (Fig. 1H).

Re: We found that 53BP1 is pulled down by ASF1a or ASF1b, although it could not be detected by mass spectrometry (Fig. 1c), suggesting that the RIF1-ASF1 complex associates with 53BP1, perhaps weakly.

We examined these interactions using soluble and chromatin fractions. Although the abundance of FLAG-ASF1a in the chromatin fraction was lower than that in the soluble fraction, 53BP1 and RIF1 accumulated at greater levels in the IP of FLAG-ASF1a from the chromatin fraction (Supplementary Fig. 2a). Therefore, the interactions of ASF1a with RIF1 and 53BP1 in the chromatin fraction were stronger than those in the soluble fraction, consistent with this reviewer's point.

We performed the IP in Fig. 1C with or without bleomycin treatment as suggested. The abundance of 53BP1 and RIF1 was increased in the Flag-ASF1a and ASF1b IPs (Fig. 1c).

We added these results in the new manuscript.

The interaction of endogenous ASF1B with RIF1 may be very weak, and we therefore did not detect an obvious band of ASF1B in the RIF1 pull-down results shown in Fig. 1B. Overexpression of exogenous GFP-ASF1B may enhance its interaction with RIF1, as shown in the previous version of Fig. 1H.

- Supp. Fig. 2: it is surprising not to detect H3 in HIRA and CAF-1 pull-downs.

Re: We repeated the experiment many times and could not detect H3 in the HIRA and CAF-1 pull-downs. One possibility is that the detergents in our lysis buffer affected their interactions.

- The authors should provide a control WB for RIF1 KO (Fig.1), and should control the effect of 53BP1 and RIF1 KO on ASF1 total levels since an upregulation of ASF1 could mask ASF1 local enrichment on laser tracks (Fig. 1F-G). Similarly, the KO of rif1, brca1, and shld2 genes in DT40 cells should be verified (Supp. Fig. 1). In Fig. 6, the authors should control the effect of 53BP1-, RIF1- and ASF1-KO on the total levels of Suv39, and should assess whether the depletion of Suv39 impacts the total levels of BRCA1 and RPA. Indeed, knocking-down a transcriptional repressor may affect the expression of many genes.

Re: We thank the reviewer for these suggestions. We provided a control WB for RIF1 KO cells (Supplementary Fig. 3a). ASF1a/b levels were not changed after 53BP1 and RIF1 knockout (Supplementary Fig. 3a). Knockout of the rif1, brca1, and shld2 genes in DT40 cells was also verified (Supplementary Fig.1c). 53BP1 knockout did not significantly change the protein level of SUV39H1/2. SUV39H1, but not SUV39H2, was slightly reduced in RIF1- or ASF1-deficient cells (Supplementary Fig. 3a). Depletion of SUV39H1/2 did not significantly impact the total levels of BRCA1 and RPA (Fig. 6c). We included these results in the new manuscript.

- Fig. 6E: siSUV39H1/2 leads to a global decrease in H3K9me3, as expected. Therefore, it is difficult to conclude about a local decrease of this mark at DSBs.

Re: We agree with this reviewer that it is difficult to exclude whether the local decrease in H3K9me3 at DSBs was due to a global decrease in the abundance of this mark. This result is only consistent with our hypothesis. We changed our statement in the new manuscript and indicated that these results are only consistent with our point.

- Suppl. Fig. 1: the effects of shieldin and rif1 loss are compared in chicken DT40 cells. Could the authors state whether paralogs of shld2 exists in chicken, which might explain why the shieldin2 KO is not as effective as the RIF1 KO in rescuing the BRCA1 deficiency in response to PARPi.

Re: Yes, one possibility is that a paralogs of shld2 exists in chicken to explain the data. However, we did not find it by searching the chicken genome on the NCBI database. Therefore, we speculated that a parallel pathway of Shieldin exists. Our data show that ASF1 and Shieldin act at two parallel steps downstream of RIF1 to promote NHEJ.

- Line 179: the authors should rephrase the sentence stating that ASF1A recruitment to the core

region was not significantly affected by the disruption of 53BP1 or RIF1 because Fig. 2I shows an increase of at least 2-fold of ASF1A recruitment to DSB-proximal regions in both mutants compared to control cells.

Re: We thank this reviewer for his/her careful reading. We changed this statement to “Recruitment of ASF1a and ASF1b to the core regions was not reduced by disruption of 53BP1 or RIF1 (Fig. 2H-J)” in the new manuscript.

- Laser micro-irradiation has long been recognized as a useful tool to assess the spatial distribution of repair factor recruitment to DSBs (PMID: 16618811). Thus, the HIRDC method is not strikingly novel. Moreover, the authors should specify in the method section which laser wavelength is used for laser micro-irradiation experiments. Surprisingly, it seems that the standard and high-energy laser experiments both use 65% laser output, while a higher laser output would be expected for the high-energy conditions.

Re: We indicated the laser wavelength (365 nm) in the methods section of the new manuscript as suggested. Micro-Point generates micro-irradiated dots infrequently along a line (under default setting). In the HIRDC method, micro-irradiated dots are densely distributed in a preset circular region (approximately 1 μ m). Although 65% laser output was used in both experiments, the radiation dose of the DNA in the HIRDC will be the superimposed result of close irradiated-dots. Therefore, in the HIRDC assay, the final dose of micro-irradiation is much higher.

- Fig. 2J, the minimal number of cells scored per condition should be indicated. Regarding statistics, one-way ANOVA should be used instead of t-tests when comparing more than two samples. This applies to all statistical tests in this study. Moreover, it would be helpful to include statistics in Fig. 1I, and in Fig. 6L (brca1-/- vs brca1-/-suv39h-/- survival to Olaparib).

Re: At least 50 cells were scored for every sample per condition. We included this information in the figure legends as suggested.

We used one-way ANOVA instead of t-tests when comparing more than two samples as suggested. Statistics in Fig. 1i and Fig. 6k (old Fig. 6L) were added as suggested.

- The authors should explain why they analyzed two HR markers, RPA32 and RAD51, in different cell lines, chicken DT40 and human U2OS (Fig. 3K, Suppl. Fig. 5A).

Re: The RPA32 foci in DT40 cells were quite weak and difficult to quantify. It was necessary to use another cell line (U2OS) to assess the RPA32 foci.

- The authors should justify the focus on HP1gamma (as opposed to alpha and beta).

Re: All of HP1 isoforms, alpha, beta and gamma were recruited to DSB sites by 53BP1, RIF1 and ASF1 (Supplementary Fig. 10a-f). Because previous studies showed that HP1 alpha and beta tend to promote HR, while gamma tends to stimulate NHEJ (PMID: 25818296; PMID: 23287531), we focused on HP1gamma.

- Fig. 4C: the authors could specify that they measure mean intensities (bottom panels), which are insensitive to changes in area (top panel).

Re: We examined the H3K9me3 signal using CHIP and obtained similar results (Fig. 5d). Therefore, in this system, mean intensities do not seem to be very sensitive to changes in area.

- Final model in Fig. 7: In order to simplify the message, we recommend that the authors only show the panel A.

Re: We removed panel B as suggested.

Discussion points:

- Both ASF1 and RIF1 are conserved in budding yeast, thus, whether the role of ASF1 in regulating chromatin compaction and resection at DSBs is evolutionary conserved could be discussed.

- Line 365: What would support the assumption that RIF1 may have a histone chaperone activity? Does RIF1 harbor an acidic domain that may contribute to histone binding?

- Lines 447-448: Note that the combination of HDAC inhibitors and PARP inhibitors has already been tested and shown efficacy for the treatment of breast cancer (PMID: 25026298) and glioblastoma for example (PMID: 26794465).

Re: We could not find the B domain in budding yeast RIF1. Therefore, this complex is unlikely to be conserved in yeast.

RIF1 harbors a conserved acidic domain close to the B domain. It remains to be determined whether this acidic domain is required for regulating chromatin compaction and suppressing resection at DSBs. In vitro biochemical assays may be able to verify whether RIF1 has histone chaperone activity.

HDAC1/2 was reported to promote NHEJ (PMID: 20802485), consistent with our study. Inhibition of HDAC promotes HR and promotes resistance to PARP inhibitors. Therefore, HDAC inhibitors may act at other pathways to confer PARP inhibitor sensitivity in breast cancer and glioblastoma. We removed the discussion material about HDAC1/2.

We added relevant discussions material in the new manuscript.

Typographical errors:

- Line 76, the title of the first result section needs rephrasing.

- Line 325, "meditated" instead of "mediated".

- Line 370, a verb is missing at the beginning of the sentence.

- Lines 386 and 433, replace "ensure" by "ensue".

- Line 445, replace "HADC" by "HDAC".

- Fig. 3E, the red curve should be labeled *asf1a*^{-/+}.

Re: We thank this reviewer for his/her careful reading. We corrected these typographical errors.

REVIEWER COMMENTS

Reviewer #1 (Remarks to the Author):

The authors have satisfactorily answered all my concerns and I would recommend the manuscript for publication in Nature Communications.

Reviewer #2 (Remarks to the Author):

The authors have provided a considerable revision and improved the MS. Please find my comments to previously raised points 1-6 below. (1) Re: the apparent discrepancy between the effects of Rif1_AAD (ASF1-interaction mutant) and ASF1_EDAA (RIF1-interaction mutant), with the former promoting significant rescue of rif1-/-etoposide resistance and the latter unable to provide any rescue in this setting; this data suggests an ASF1_EDAA defect in DNA break repair unrelated to its RIF1 interaction. It may be reassuring to see that Rif1_AAD and ASF1_EDAA show a similar reduction in the now added random plasmid integration assay. These new results do provide added support that RIF1-ASF1 interactions may promote NHEJ reactions. However, random DNA integration is an extremely rare event and difficult to relate to a DNA damage repair assay. Given that etoposide is still the only DNA double-strand break-inducing agent used in the MS to directly compare the effects of Rif1_AAD and ASF1_EDAA on cell survival, and given the disparity in outcome, there remains a need to discuss the possibility that ASF1 contributes to the repair of exogenous DNA double-strand breaks by NHEJ mainly through a pathway unrelated to the direct RIF1 physical interaction elaborated in this MS. In fact, the contribution of RIF1 B domain – ASF1 interactions may be rather limited (casting some doubt over the idea of a 53BP1-RIF1-ASF1 axis of functional significance for DSB repair), at least for NHEJ repair of induced DNA double-strand breaks. As mentioned previously, this should be discussed in light of an alternative model for ASF1 in DNA double-strand break repair by NHEJ published in ref. 71 (ref. 68 in previous version of the MS). The effects with etoposide herein might suggest that an ASF1-dependent repair defect results mainly from defective ATM signalling, as previously proposed? Thus, the added comment in the revised MS regarding the possibility that RIF1_AAD might “partially resist etoposide through its interaction with shieldin” rather misses the point and does not explain why ASF1 mutations (including EDAA) are so much more dramatic in the presence of etoposide than the RIF1_AAD mutation. This goes right to the heart of the potential novelty of the current MS, and the question whether a significant new molecular mechanism promoting the repair of double-strand breaks by NHEJ with effects for cell survival, distinct from previously published work (e.g., ref. 71), has been elucidated.

(2) This point has been answered.

(3) This point has been adequately addressed.

(4) Point has been addressed.

(5) This has been improved in the revised version of the MS.

(6) This has been improved; significance for cancer in the last para of discussion (I believe unchanged from the previous version of the MS) remains doubtful given the unresolved discrepancies that apparently exist in DT40 herein vs. published findings in mammalian system. It would be good to make clear that this is a mechanistic extrapolation with the caveat of potential differences between DT40 and human cells in the context of the last para.

Reviewer #3 (Remarks to the Author):

The authors have very satisfactorily and thoroughly addressed the reviewers' comments. Their revised manuscript is strengthened by a number of additional experiments and controls. They provide solid evidence that ASF1 and RIF1 function in the same pathway to trigger heterochromatinization at DSBs, antagonize BRCA1-dependent HR and promote NHEJ. Only a few minor issues require attention before this manuscript can be accepted for publication:

The authors should specify in the method section:
which antibody is used to detect ASF1A in IF (Suppl. Fig. 3b-d),
which primers are used for the H3K9me3 ChIP-qPCR (Fig. 5d, this piece of information is difficult to find if present only in the legend of Fig. 5a),
the siRNA sequence for CAF-1 p150,
the procedure for RT-PCR and the primers used to detect CAF-1 p60, p150
they should also include HCT116 in the cell culture paragraph.

Fig. 7: Please provide a more extensive legend for the model presented in Fig. 7 explaining the role of Suv39h1/2 in H3K9 trimethylation and of the different nucleases in resection. The positioning of these molecular events with respect to the DSB end suggests a recruitment of ASF1 distal to the break and a heterochromatinization closer to the break, which is misleading.

REVIEWER COMMENTS

Reviewer #1 (Remarks to the Author):

The authors have satisfactorily answered all my concerns and I would recommend the manuscript for publication in Nature Communications.

Re: We thank this reviewer for his/her recommendation.

Reviewer #2 (Remarks to the Author):

The authors have provided a considerable revision and improved the MS. Please find my comments to previously raised points 1-6 below. (1) Re: the apparent discrepancy between the effects of Rif1_AAD (ASF1-interaction mutant) and ASF1_EDAA (RIF1-interaction mutant), with the former promoting significant rescue of *rif1*^{-/-}-etoposide resistance and the latter unable to provide any rescue in this setting; this data suggests an ASF1_EDAA defect in DNA break repair unrelated to its RIF1 interaction. It may be reassuring to see that Rif1_AAD and ASF1_EDAA show a similar reduction in the now added random plasmid integration assay. These new results do provide added support that RIF1-ASF1 interactions may promote NHEJ reactions. However, random DNA integration is an extremely rare event and difficult to relate to a DNA damage repair assay. Given that etoposide is still the only DNA double-strand break-inducing agent used in the MS to directly compare the effects of Rif1_AAD and ASF1_EDAA on cell survival, and given the disparity in outcome, there remains a need to discuss the possibility that ASF1 contributes to the repair of exogenous DNA double-strand breaks by NHEJ mainly through a pathway unrelated to the direct RIF1 physical interaction elaborated in this MS. In fact, the contribution of RIF1 B domain – ASF1 interactions may be rather limited (casting some doubt over the idea of a 53BP1-RIF1-ASF1 axis of functional significance for DSB repair), at least for NHEJ repair of induced DNA double-strand breaks. As mentioned previously, this should be discussed in light of an alternative model for ASF1 in DNA double-strand break repair by NHEJ published in ref. 71 (ref. 68 in previous version of the MS). The effects with etoposide herein might suggest that an ASF1-dependent repair defect results mainly from defective ATM signalling, as previously proposed? Thus, the added comment in the revised MS regarding the possibility that RIF1_AAD might “partially resist etoposide through its interaction with shieldin” rather misses the point and does not explain why ASF1 mutations (including EDAA) are so much more dramatic in the presence of etoposide than the RIF1_AAD mutation. This goes right to the heart of the potential novelty of the current MS, and the question whether a significant new molecular mechanism promoting the repair of double-strand breaks by NHEJ with effects for cell survival, distinct from previously published work (e.g., ref. 71), has been elucidated.

Re: The major concern of this reviewer is the discrepancy between the effects of Rif1_AAD and ASF1a_EDAA in repair of etoposide-induced DSBs: (1) RIF1_AAD can still partially rescue the etoposide sensitivity of *rif1*^{-/-} cells, while ASF1a_EDAA fully loses this function; (2) ASF1a_EDAA is “much more dramatic in the presence of etoposide than the RIF1_AAD mutation”. We agree with this reviewer that the added comment in the revised MS regarding the possibility that RIF1_AAD might partially resist etoposide through its interaction with shieldin can't explain this discrepancy well.

We thank this reviewer for his/her deep thinking and propose a new explanation after reading more literature and thinking more deeply.

Etoposide-induced DSBs are repaired not only by NHEJ, but also by other pathways including HR (PMID: 20088963). Moreover, etoposide induces DNA-TOP2 adducts, which need additional processing and are able to block replication (PMID: 27376333). It has been reported that RIF1 is also required for cellular response to replication stress (PMID: 19948482; PMID: 20711169) and RIF1 knockout mice or cells are much more severe than the 53BP1 deficient ones (PMID: 19948482; PMID: 12640136). Therefore, RIF1 may have multiple functions in resisting etoposide, and its interaction with ASF1 is only required for NHEJ repair, which explains why RIF1_AAD can still partially rescue the etoposide sensitivity of RIF1 knockout cells. On the other hand, perhaps, because we used different scale ranges of Y-axis in Fig. 3c and 3e, this reviewer was misled to think that ASF1a_EDAA is much more dramatic in the presence of etoposide than the RIF1_AAD mutation. In fact, if we place the curves of *rif1*^{-/-}+RIF1_AAD cells and ASF1a deficient cells in the same plot (see Figure below), ASF1a deficient cells are only weakly more sensitive to etoposide than *rif1*^{-/-}+RIF1_AAD cells. ASF1 is required for both NHEJ and HR through its interaction with RIF1 and CAF-1, respectively (this paper and PMID: 29478807). Both NHEJ and HR are required to repair etoposide-induced DSBs. Therefore, ASF1a_EDAA, which loses both its interactions with RIF1 and CAF-1, is more sensitive to etoposide than the RIF1_AAD mutation, possibly due to additional defect of ASF1 in HR. We added these explanations in the revised manuscript.

Figure legend: Etoposide sensitivity of various groups of DT40 cells. This figure is a combination of parts of Fig.3a and 3c. The experiments of Fig.3a and 3c were carried out at the same time.

In addition, we carried out complementary experiments with regard to PARP inhibitor (PARPi) sensitivity in *BRCA1*^{-/-}*RIF1*^{-/-} double knockout cells. Likely, this reviewer ignored this result because it was placed not in Fig. 3, but in Fig. 4h. Conferring PARPi sensitivity in BRCA1-mutated cells is a very specific function of the 53BP1 pathway through

protecting the broken ends and promoting NHEJ (PMID: 30948458). It was proposed that NHEJ-mediated chromosome aberrations induce cell death of BRCA1-deficient cells in the presence of PARPi (PMID: 20362325). All well-known factors of the 53BP1 pathway, including 53BP1, RIF1, REV7 and Shld1/2/3, have been reported to display this function (PMID: 20362325; PMID: 20453858; PMID: 23333306; PMID: 23333305; PMID: 23306439; PMID: 23306437; PMID: 25799992; PMID: 25799990; PMID: 30046110; PMID: 30022168; PMID: 29789392; PMID: 30154076; PMID: 30022119; PMID: 30254264; PMID: 29656893). Our data showed that the wild-type RIF1 almost fully rescued the PARPi-sensitivity of the *BRCA1*^{-/-}*RIF1*^{-/-} cells, while RIF1_AAD almost totally lost this function (Fig. 4h). Therefore, the ASF1-RIF1 interaction is important for their roles in antagonizing BRCA1 and promoting NHEJ.

Moreover, complementary experiments in random plasmid integration assay further confirmed this conclusion. Random integration together with telomere fusion assay and class-switch recombination (CSR) assay are well established report systems for NHEJ efficiency. Although all of these (random integration, telomere fusion and CSR) are rare events in most cells and are difficult to directly relate to a DNA damage repair assay, they are widely used for examining NHEJ efficiency, including the studies of RIF1 (PMID: 23333306; PMID: 23333305; PMID: 23306439; PMID: 23306437), REV7 (PMID: 25799992; PMID: 25799990) and Shld1-3 (PMID: 30046110; PMID: 30022168; PMID: 29789392; PMID: 30154076; PMID: 30254264; PMID: 29656893). Therefore, our data of random plasmid integration assay should also be regarded as important evidence. Additionally, PARPi-sensitivity is a DNA damage repair assay.

Finally, the defect of ASF1a deficient cells in resisting etoposide is unlikely due to defective ATM signaling for the following reasons. First, ASF1a_V94R, which fully retained its function in ATM signaling (ref.71, PMID: 28943310), totally lost its function in resisting etoposide in DT40 cells (Fig. 3e). Therefore, defective ATM signaling is unlikely the cause that induces etoposide-sensitivity in DT40 cells when ASF1a is depleted. Second, ASF1-dependent ATM signaling is independent of its histone chaperone activity as reported in ref.71 (PMID: 28943310), and thus presumably, it does not require the interaction of ASF1 with CAF-1 in human cells. In other words, ASF1a_EDAA should retain its function on ATM signaling in human cells although they did not examine it (ref.71, PMID: 28943310). However, ASF1a_EDAA fully lost its function in resisting etoposide in DT40 cells (Fig. 3e). Third, ATM is required for cell to resist PARPi (PMID: 18832051; PMID: 18388863; PMID: 16912188; PMID: 22416035; PMID: 30622252). ATM aberration in tumors has been selected as one of the biomarkers for clinical studies of PARPi treatment (PMID: 28302823; PMID: 32367009; PMID: 32117762). We also found that ATM inhibition re-sensitized the *brca1*^{-/-}*53bp1*^{-/-} cells to PARPi (see below). In contrast, depletion of ASF1a resulted in PARPi resistance of *BRCA1*^{-/-} DT40 cells and HCT116 cells (Fig. 4a-c). Therefore, depletion of ASF1a is unlikely to cause serious defects of ATM signaling in DT40 cells and some human cells.

Legend: Olaparib sensitivity of *brca1*^{-/-}*53bp1*^{-/-} DT40 cells in presence of ATM inhibitor (KU-55933, 5 μ M).

(2) This point has been answered.

(3) This point has been adequately addressed.

(4) Point has been addressed.

(5) This has been improved in the revised version of the MS.

(6) This has been improved; significance for cancer in the last para of discussion (I believe unchanged from the previous version of the MS) remains doubtful given the unresolved discrepancies that apparently exist in DT40 herein vs. published findings in mammalian system. It would be good to make clear that this is a mechanistic extrapolation with the caveat of potential differences between DT40 and human cells in the context of the last para.

Re: We thank this reviewer for his/her suggestion. To remind the potential differences between DT40 and human cells, we modified the last sentence as “If our conclusions, which are mainly acquired from chicken DT40 cells, are applied to human tumor cells, future investigations should assess whether such epigenetic enzymes are biomarkers that could indicate the potential responses of cancers to PARPi therapy.”

Reviewer #3 (Remarks to the Author):

The authors have very satisfactorily and thoroughly addressed the reviewers’ comments. Their revised manuscript is strengthened by a number of additional experiments and controls. They provide solid evidence that ASF1 and RIF1 function in the same pathway to trigger heterochromatinization at DSBs, antagonize BRCA1-dependent HR and promote NHEJ. Only a few minor issues require attention before this manuscript can be accepted for publication:

Re: We thank this reviewer for his/her recommendation.

The authors should specify in the method section:
which antibody is used to detect ASF1A in IF (Suppl. Fig. 3b-d),

which primers are used for the H3K9me3 CHIP-qPCR (Fig. 5d, this piece of information is difficult to find if present only in the legend of Fig. 5a),

the siRNA sequence for CAF-1 p150,

the procedure for RT-PCR and the primers used to detect CAF-1 p60, p150

they should also include HCT116 in the cell culture paragraph.

Re: We thank this reviewer for his/her careful reading. We added this information in the new manuscript.

Fig. 7: Please provide a more extensive legend for the model presented in Fig. 7 explaining the role of Suv39h1/2 in H3K9 trimethylation and of the different nucleases in resection. The positioning of these molecular events with respect to the DSB end suggests a recruitment of ASF1 distal to the break and a heterochromatinization closer to the break, which is misleading.

Re: We thank this reviewer for his/her suggestion. We added a more extensive legend to the model in the new manuscript.

REVIEWERS' COMMENTS

Reviewer #2 (Remarks to the Author):

The authors have considered major point (1) and have now commented on the disparity of etoposide phenotypes seen with Rif1_AAD (ASF1 interaction mutant) and ASF1_EDAA (RIF1 interaction mutant). This remains a somewhat puzzling result, but in absence of complementary assays with other DNA double-strand break-causing treatments, the authors offer some potential explanations for the findings, which now allows for the possibility that ASF1 contributes to the repair of exogenous DNA double strand breaks considerably through a pathway unrelated to the direct RIF1 physical interaction elaborated in this MS. This reflects the current depth of the data on this point more faithfully. Point (6), the other remaining issue, has also been addressed in the text.

REVIEWERS' COMMENTS

Reviewer #2 (Remarks to the Author):

The authors have considered major point (1) and have now commented on the disparity of etoposide phenotypes seen with Rif1_AAD (ASF1 interaction mutant) and ASF1_EDAA (RIF1 interaction mutant). This remains a somewhat puzzling result, but in absence of complementary assays with other DNA double-strand break-causing treatments, the authors offer some potential explanations for the findings, which now allows for the possibility that ASF1 contributes to the repair of exogenous DNA double strand breaks considerably through a pathway unrelated to the direct RIF1 physical interaction elaborated in this MS. This reflects the current depth of the data on this point more faithfully. Point (6), the other remaining issue, has also been addressed in the text.

Re: We thank this reviewer for his/her comments and recommendation.